# Under the knife: Unfavorable perceptions of women who seek plastic surgery

**Sarah Bonell**◉*, **Sean C. Murphy, Scott Griffiths**

Melbourne School of Psychological Sciences, The University of Melbourne, Melbourne, Victoria, Australia

* sbonell@student.unimelb.edu.au

## Abstract

Plastic surgery is growing in popularity. Despite this, there has been little exploration to date regarding the psychosocial consequences of seeking plastic surgery. Our study investigated how women seeking plastic surgery are perceived by others. We presented a random sample of 985 adults (men = 54%, $M_{age}$ = 35.84 years, $SD_{age}$ = 10.59) recruited via Amazon's Mechanical Turk with a series of experimental stimuli consisting of a photographed woman (attractive versus unattractive) and a vignette describing an activity she plans to engage in (plastic surgery versus control activity). Participants rated stimuli on perceived warmth, competence, morality, and humanness. We ran linear mixed-effect models to assess all study hypotheses. There was a negative plastic surgery effect; that is, women seeking plastic surgery were perceived less favorably than those planning to complete control activities across all outcome variables (warmth, competence, morality, and humanness). These relationships were moderated by physical attractiveness; while attractive women planning to undergo plastic surgery were perceived less favorably than attractive women planning to engage in control activities, perceptions of unattractive individuals remained unchanged by plastic surgery status. We theorized that empathy toward unattractive women seeking plastic surgery mitigated the negative plastic surgery effect for these women. In sum, our results suggest that perceptions of attractive women are worsened when these women decide to seek cosmetic surgery. Perceptions of warmth and competence have implications for an individual's self-esteem and interpersonal relationships, while perceptions of morality and humanness can impact an individual's ability to fulfil their psychological needs. As such, we concluded that attractive women seeking plastic surgery are potentially subject to experience negative psychosocial outcomes. Future research ought to examine whether perceptions and outcomes differ for women seeking reconstructive plastic surgery (versus cosmetic plastic surgery) and whether they differ across different types of surgeries (i.e. face versus body).

## Introduction

The number of individuals electing to undergo plastic surgery has steadily increased in Western societies since its introduction following the First World War [1]. Plastic surgery is particularly popular among women, who account for approximately 87% of all plastic surgery

**Data Availability Statement:** All data files are available from the Open Science Framework database (https://tinyurl.com/ska2qv9).

**Funding:** SG is supported by a National Health and Medical Research Council Early Career Fellowship

(grant number: 1121538; funder website: https://www.nhmrc.gov.au). The funders had no role in study design, data collection and analysis, decision to publish, or preparation of the manuscript.

**Competing interests:** The authors have declared that no competing interests exist.

recipients [2]. Today, nearly 15 million plastic surgeries per year are performed on women in the US alone–a 169% increase over the past 20 years [3]. Further, plastic surgery rates increased by more than 20% worldwide between the years 2015 and 2019 [2]. Research suggests that plastic surgery may benefit recipients both physically and psychologically; complication rates are relatively low, and recipients typically report feeling satisfied with results [4–6]. Recipients also report both improved psychological wellbeing and greater appearance satisfaction up to five years post-surgery [6–8]. That said, there has been little exploration to date regarding the potential social consequences plastic surgery recipients may face. Specifically, it remains unclear how the decision to undergo plastic surgery is perceived by other members of society.

### The negative plastic surgery effect: Perceptions of plastic surgery recipients

There is some qualitative evidence to suggest that plastic surgery recipients are perceived negatively by others. Saxena [9] explored recipients' lived experiences of plastic surgery stigmatization by conducting qualitative interviews with twenty women who had undergone cosmetic breast augmentation. She found that these women reported feeling stigmatized in that they were perceived by others to be psychologically unstable and insecure. For example, one respondent commented that "when you confess they're fake, people think about you as unstable, that you don't have any confidence in yourself and that's why you have to get them". Similarly, Bonell and colleagues [10] investigated experiences of stigma among 15 Australian women who had undergone plastic surgery. They found that women reported being stigmatized because others perceived them to be mentally unwell or unnatural (e.g., a husband commenting "I don't want you to turn into this, you know, fake kind of thing.") Finally, Ricciardelli and Clow [11] investigated plastic surgery attitudes among 103 Canadian men. They found that these men perceived plastic surgery recipients to be vain and lazy (e.g., "I think cosmetic surgery is a lazy way out"). From a quantitative perspective, existing literature has shown that cross-cultural approval ratings for plastic surgery are low [12, 13]. In one study, Delinsky found that 302 undergraduate students from the US attributed negative personality traits to recipients of plastic surgery (i.e., materialistic, self-conscious, and perfectionistic) and considered them to have poor mental wellbeing [12]. Similarly, Tam and colleagues found across Hong Kongese, Japanese, and American samples that plastic surgery was broadly considered unacceptable, and that participants ascribed negative characteristics to, and refused to have social relationships with, plastic surgery recipients [13]. In sum, previous literature has established that there are prevailing hostile attitudes toward plastic surgery and its recipients. However, this phenomenon is yet to be explored experimentally.

There is also anecdotal evidence to suggest that plastic surgery is stigmatized. For instance, tabloid news articles [14] and social media accounts (e.g., @celebface, 1.3m followers on *Instagram*) regularly publish exposés "outing" celebrities who have had plastic surgery, signifying that this behavior is considered embarrassing and/or shameful. That said, perceptions of plastic surgery are nuanced on a global scale; in some societies, plastic surgery is engrained as a part of cultural identity. In Brazil, for example, plastic surgery is celebrated as a means through which women might advance both their personal and professional lives (i.e. because education is limited, appearance has become an important source of power for Brazilian women) [13, 15, 16]. As a result, Brazilian perceptions of plastic surgery are overwhelmingly positive [17]. In sum, perceptions of plastic surgery are complex and not yet fully understood. However, it seems that in some contexts plastic surgery recipients are indeed perceived unfavorably (i.e. there is a negative plastic surgery effect). These perceptions might be considered a 'horn' or 'negative halo' effect–a cognitive bias whereby perceptions of an individual are unduly influenced by a single negative trait [18, 19].

Importantly, there has been no research to date in which perceptions of women *planning to undergo* plastic surgery have been explicitly explored (i.e. focus has always been on perceptions of women who have *already undergone* plastic surgery). We posit that exploring plastic surgery as an intended action (versus a completed action) enables researchers to better assess negative perceptions that pertain exclusively to plastic surgery itself, rather than its associated outcomes (i.e. how recipients *look* after surgery). In other words, studying perceptions of women planning to have plastic surgery allows us to assess shifts in perception regarding recipient character (e.g., "I don't condone plastic surgery because it is immoral") as opposed to regarding recipient appearance (e.g., "I don't condone plastic surgery because I think it makes women look unappealing"). As such, our study aims to establish exactly how a woman's decision to undergo plastic surgery shapes others' perceptions of her, irrespective of her surgical outcomes. This knowledge will help contribute to our growing understanding of how society perceives plastic surgery recipients (i.e. whether they are stigmatized).

**The negative impact of stigma.** Stigmatized groups face considerable challenges. For example, mental health stigma in the workplace can increase employee's work-related stress and reduce longevity of employment [19]. Similarly, addiction stigma can isolate users from both their social networks and support services [20]. Finally, stigmatized sexual minorities are subject to intrusive thoughts and physical symptoms (e.g., diarrhea, faintness, cold, or cough) [21]. Thus, there is reason to believe that if plastic surgery is indeed stigmatized, this will adversely impact recipients. Therefore, it is important that we understand whether women who undergo plastic surgery are indeed stigmatized.

## Physical attractiveness and the negative plastic surgery effect

It is imperative that we establish not only whether there is a negative plastic surgery effect, but also whether the characteristics of women seeking plastic surgery influence the degree to which they are subject to this effect. For example, existing literature has demonstrated that an individual's physical attractiveness–by which we mean their average attractiveness score as rated by others–affects how others perceive them. Typically, being attractive is associated with superior perceptions; for instance, attractive people are assumed to be warmer and more capable than unattractive people [22–24]. There is also some evidence to suggest that attractive individuals are considered morally superior. For example, attractive people are less likely to be convicted of crimes than unattractive people and receive less severe sentences upon conviction [25, 26]. In certain contexts, however, being attractive can prove detrimental. For instance, because they are assumed to be more competent and capable of managing their own circumstances, attractive children who face hardship are subject to less empathy from adults than unattractive children in identical scenarios, meaning that they are also less likely to receive adult support [27]. As such, we might conclude that attractiveness, while typically beneficial, is a double-edged sword.

In the present study, we therefore propose that the attractiveness of women intending to undergo plastic surgery might influence the presence or magnitude of the negative plastic surgery effect. Put simply, because attractive and unattractive people's abilities, motivations, and personalities are considered to differ systematically across a variety of circumstances, we believe that attractive and unattractive plastic surgery recipients might be perceived differently by members of society. Given that there is no research to date examining how attractiveness might influence person perception specifically in appearance-enhancement contexts, however, we can only speculate regarding the directionality of the influence that recipient attractiveness might have on the negative plastic surgery effect.

## For whom might the negative plastic surgery effect be strongest?

As well as contextualizing for which recipients the negative plastic surgery effect is greatest, we also feel it important to contextualize the kind of individual who is most likely to ascribe the negative plastic surgery effect to recipients. For the purposes of this study, we explore justice sensitivity and disgust sensitivity as two potential moderators for perceptions of women seeking plastic surgery. In other words, we examine whether individuals more sensitive to injustices and/or disgust are also more likely to condemn plastic surgery.

**Justice sensitivity.** Justice sensitivity describes the extent to which one feels negatively toward perceived injustices (e.g., when someone gets something they don't deserve), while the 'beauty-as-currency' hypothesis describes the theory that attractiveness is an accruable social currency that, much like wealth or social status, grants access to certain privileges [28–30]. If beauty is indeed a form of social currency, it would follow that those higher in justice sensitivity might condemn plastic surgery because it allows recipients to profit from 'unearned' appearance enhancements. For instance, previous literature has demonstrated there is an 'effort bias' when perceptions of individuals who are fat are formed; those who diet or exercise to lose weight are perceived more favorably than those who undergo surgery to lose weight [31, 32]. In the context of plastic surgery, those higher on justice sensitivity might find it problematic that plastic surgery recipients obtain the benefits that come with being beautiful (e.g., status, privilege) without doing any of the 'work' usually required to obtain these benefits (e.g., arduous dieting and exercising) [11, 28]. Furthermore, those higher on justice sensitivity may resent that there is class-based privilege associated with plastic surgery accessibility (i.e. financial barriers preclude some individuals from having surgery). Taken together, we believe that justice sensitivity might moderate perceptions of plastic surgery recipients, such that those more sensitive to injustices will perceive women intending to undergo plastic surgery less favorably.

**Disgust sensitivity.** Existing literature has shown that disgust sensitivity (i.e. one's predisposition toward experiencing disgust) plays an integral role in appearance-related person perception. Disgust is a visceral reaction that evolutionarily developed as a survival mechanism. Because individuals are inherently motivated to avoid that which they find disgusting, disgust plays a role in the minimization of exposure to disease and pathogen threat by inducing avoidance toward unfamiliar stimuli [33–36]. Existing literature has demonstrated that individuals with non-normative bodies (i.e. those that don't align with dominant societal perceptions of how bodies ought to look or be) elicit disgust reactions, likely because their unfamiliarly alerts our behavioral immune system that a pathogen threat is present [33]. Further, past studies have shown that an individual's level of disgust sensitivity is predictive of their stigmatization toward individuals with non-normative bodies (e.g., people with disabilities or people who are fat) [33, 37]. We therefore posit that those more sensitive to disgust might also express greater plastic surgery stigmatization, given that recipients are also planning to acquire a kind of non-normative body (i.e. one that is is stigmatized because surgically enhanced bodies do not align with perceptions of what bodies ought to be).

## Measuring the negative plastic surgery effect

First proposed in 2002, the Stereotype Content Model proposes that interpersonal impression formation is best understood as a product of two fundamental dimensions: warmth and competence [38]. Put simply, the way we feel about others is said to depend on whether we consider them to be warm (e.g., friendly, trustworthy) and/or competent (e.g., capable, assertive). The model is based in evolutionarily theory; individuals are innately predisposed to assess both a stranger's intent to either hurt or help them (warmth) and that stranger's capacity to act

on said intention (competence) [38–40]. While the Stereotype Content Model is still widely used in psychological literature, emerging evidence suggests that the warmth dimension of the model subsumes two independent measures of person perception: warmth and morality. Consequently, it has been proposed that distinctions need be made between traits that reflect warmth and those that reflect morality, as well as those that reflect both [39–43]. Further, a relatively less explored dimension of person perception is the attribution of humanness to other individuals. Existing literature has demonstrated that women who engage with beautification are perceived to be less human [44]. As such, we are likely to see decreased perceptions of humanness among plastic surgery recipients. It is important that we examine humanness in conjunction with the Stereotype Content Model because perceptions of humanness directly influence interpersonal relationships [45, 46]. For instance, people who are dehumanized are more often victims of objectification and aggression, and receive less empathy from others [46]. Hence, it is imperative that we understand whether plastic surgery recipients are dehumanized. In sum, we will examine the social consequences of undergoing plastic surgery by examining how women who seek plastic surgery are perceived across four domains: warmth, competence, morality, and humanness.

## Study aims and hypotheses

We aimed to examine whether perceptions of women who seek plastic surgery systematically differ from perceptions of women who do not. Our primary hypothesis was that (1) women seeking plastic surgery would be considered less warm, competent, moral, and human than those who are not (i.e. there would be a negative plastic surgery effect). We also hypothesized that (2) this relationship would be moderated by both justice sensitivity and disgust sensitivity, such that the negative plastic surgery effect would be greater for those higher in justice sensitivity and disgust sensitivity. Consistent with previous research, we hypothesized that (3) unattractive women would be perceived as less warm, competent, moral, and human than attractive women [22, 23]. Finally, we proposed two secondary, exploratory hypotheses. First, we examined whether (4a) the negative plastic surgery effect was moderated by stimuli attractiveness; that is, whether attractive and unattractive women seeking plastic surgery were equally subject to the negative plastic surgery effect. Next, we explored whether (4b) the strength of this relationship differed as a function of participant justice sensitivity and/or disgust sensitivity (i.e. whether we see a three-way interaction between plastic surgery status, stimuli attractiveness, and participant justice sensitivity and/or disgust sensitivity).

## Method

### Participants

Ethics approval for the study was obtained from The University of Melbourne's Psychological Sciences Human Ethics Advisory Group (Ethics ID: 1955222) prior to study commencement. Sample size was determined prior to data analysis based on funding available to our research team at the time of data collection. Participants were 985 (men = 536; $M_{age}$ = 35.84, $SD$ = 10.59) American adults recruited via Amazon's Mechanical Turk (MTurk). The majority (69%) of participants identified their race as White ($N$ = 676; including mixed-race White), while 16% identified as Black or African American ($N$ = 155; including mixed-race Black or African American). The majority of participants identified as exclusively straight/heterosexual ($N$ = 609; 62%) or mostly straight/heterosexual ($N$ = 164; 17%). Compensation for each participant completing our 15-minute survey was US$2.33 [47]. A response inconsistency attention check was also included in our study, whereby participants were asked to identify their race at both the commencement and completion of the survey. We excluded 16 participants from the

present study for failing to consistently report their race. To elaborate, we included two items in our survey asking participants to indicate their race. In cases where participant responses did not align between these two items, 'participants' were assumed to be bots (or to simply not be paying attention) and were subsequently removed from the study [48]. For all analyses, we included participants for whom 80% or more of the relevant measures were completed.

## Materials and measures

**Stimuli.** Stimuli were each a combination of one photograph and one vignette (i.e. a short, written description). Each participant was presented with four stimuli (i.e. four photograph-vignette combinations). Photographs were taken from the Chicago Faces Database (CFD), a database that contains a series of photographed faces rated for attractiveness (i.e. 1087 participants rated the attractiveness of faces relative to other faces in the database of the same race and gender) [49]. We compiled a smaller database containing the eight most and eight least attractive (as rated by the 1087 CFD participants) White women contained in the CFD for use in the present study (i.e. 16 photographs in total). White women stimuli were chosen because we intuited that the majority of our sample would be White. Each stimulus contained one of these 16 photographs. Because each participant saw four stimuli, each participant would see four of these 16 photographs. Furthermore, we produced 13 vignettes for the present study. Of these, 12 were control vignettes that depicted a woman planning to undergo a neutral–that is, common or everyday–activity (e.g., "this woman is planning to eat a meal", "this woman is planning to buy a pet"). The remaining vignette was our plastic surgery target vignette–"this woman is planning to have plastic surgery". Again, each participant would read four of these vignettes in total; three control vignettes (of the 12 in total) and the plastic surgery vignette (vignettes can be accessed on the Open Science Framework; https://tinyurl.com/ska2qv9). Further details are outlined in the procedure.

All vignettes were assessed for their ability to induce participant affect and arousal in an MTurk pilot study ($N = 208$), as well as for their believability. Vignettes in the present study are presented in future tense (i.e. "planning to") because our pilot found that using past tense made certain vignettes seem unbelievable (i.e. "this woman recently had plastic surgery" was not believable when the stimuli photograph accompanying the vignette was unattractive). Data for the pilot study can be accessed on the Open Science Framework (https://tinyurl.com/ska2qv9).

**Justice Sensitivity Inventory—Observer Subscale.** The Justice Sensitivity Inventory–Observer Subscale (JSI-OS; Schmitt et al., 2010) is an internally consistent ($\alpha = .90$) and valid 10-item questionnaire that we used to measure the extent to which one is bothered by other people facing injustices (i.e. unfair situations) [30]. Participants rated on a 6-point scale (where $0 = $ *not at all*, $5 = $ *exactly*) the degree to which they agreed with a series of 10 statements (e.g., "I am upset when someone is treated worse than others"). Their responses were summed to calculate a total score out of 60.

At the request of a reviewer, we would like to acknowledge that while the Justice Sensitivity Inventory consists of four subscales, we ultimately felt that using solely the Observer subscale best suited the aims of our study. Namely, the Observer subscale subsumes both self-oriented and other-oriented feelings of injustice (i.e., injustices that affect both oneself and others). For example, items such as "I am upset when someone does not get a reward he/she has earned" could represent oneself as 'someone' or an external individual as 'someone'. Conversely, items on other subscales (e.g., the Victim subscale) exclusively measure feelings of injustice towards oneself. As such, we chose to use the Observer subscale because it incorporated several different possible experiences of perceived injustice and we therefore felt it more comprehensive.

**Disgust Scale—Revised.** Olatunji and colleagues' [50] Disgust Scale–Revised (DS-R) was used to measure individual differences in disgust sensitivity. The DS-R is an internally consistent ($\alpha = .79$) and valid 25-item questionnaire that contains two subscales. Subscale one asked participants to respond with either true or false (where *true* = 1, *false* = 0) to 13 disgust-related statements (e.g., "it would bother me tremendously to touch a dead body"). In this subscale, three items were negatively worded and reverse scored. Subscale two asked participants to rate on a 3-point scale (where 0 = *not*, 0.5 = *slightly*, and 1 = *very*) how disgusted they would feel in response to 12 experiences (e.g., "you see maggots on a piece of meat in an outdoor garbage pail"). Participant responses across both subscales were summed to produce a total score out of 25.

**Person perception.** Participants were asked to rate the women contained within their stimuli on the likelihood that they each possessed traits pertaining to warmth, morality, competence, and humanness. A five-point scale was used (where 1 = *extremely unlikely*, 5 = *extremely likely*).

Goodwin and colleagues [51] developed and validated a set of 32 personality traits that reflect the degree to which an individual is perceived as warm, moral, and competent. Specifically, they validated eight traits that reflect perceptions of both warmth and morality (e.g., humble, kind; $\alpha = .94$), eight that reflect perceived warmth but not morality (e.g., funny, sociable; $\alpha = .95$), eight that reflect perceived morality but not warmth (e.g., just, principled; $\alpha = .92$), and eight that reflect perceived competence (e.g., athletic, intelligent; $\alpha = .93$). In the present study, we asked participants to rate each of their stimuli on these 32 traits so that we could assess how each stimulus was perceived in terms of warmth, morality, and competence.

Haslam and colleagues [52] developed and validated a set of traits that reflect the degree to which one is considered human. In doing so, they established that being human consists of two properties: being uniquely human (UH) and displaying human nature (HN). Being UH describes possessing qualities that distinguish one from other species (i.e. qualities that humans have but that other animals do not), whereas HN describes normative human attributes (i.e. qualities that one would expect a human being to possess that might also be present in other species). Haslam and colleagues also specified that all traits can be either socially desirable or undesirable. For the present study, we assessed stimuli on four subscales each containing four terms (16 items in total): highly desirable UN traits (e.g., humble; $\alpha = .79$), highly desirable HN traits (e.g., friendly; $\alpha = .87$), undesirable UN traits (e.g., stingy; $\alpha = .82$), and undesirable HN traits (e.g., jealous; $\alpha = .58$). We operationalized our measure of *humanness* as a product of scoring highly on desirable UN and HN traits and/or scoring lowly on undesirable UN and HN traits. To rephrase, we considered 'humanness' a function of being *desirably* human.

## Procedure

Potential MTurk participants were provided with a link to an online survey hosted by Qualtrics. Upon reading details of the study and providing consent, participants completed the study's demographic measures and reported their race for the first time (attention check; see Participants section of method). Next, they completed the DS-R and JSI-OS. Following this, participants were provided with instructions that indicated they would now be "presented with photographs of women accompanied by short descriptions" and would be asked to "answer questions pertaining to each of these women and their descriptions." Each participant was presented with four stimuli in a randomized order. Two of these stimuli depicted randomly selected attractive photographs from our database, and two depicted randomly selected unattractive photographs. Three of the four photographs shown were accompanied by

randomly selected control vignettes, while one was always accompanied by the target vignette (i.e. "this woman is planning to have plastic surgery"). Assigning vignettes to photographs was randomized on a participant level; however, the target vignette was assigned to as many attractive as unattractive photographs on the study level (i.e. the same number of participants saw the target vignette assigned to an attractive photograph as they did an unattractive photograph across the whole study). When each stimulus was displayed, participants were given the prompt: "based on her photograph and description, how likely is it that the above woman is [. . .]", with the ellipsis replaced by one of 47 different terms pertaining to warmth, competence, morality, and humanness. Because the item 'humble' was present in both the warmth, morality, and competence scale [51] and the humanness scale [52], there were 47 total items for person perception instead of 48. Participants reported their race for the second time (attention check; see Participants section of method) and were then debriefed regarding the intention of the study.

## Statistical analyses

We ran linear mixed-effect models to assess all study hypotheses. These models predicted the trait rating in each trial based on fixed effects for (i) whether the photograph seen was of an attractive or unattractive woman, and (ii) whether the accompanying vignette was the target plastic surgery vignette or a control vignette. We modelled random intercepts for (i) participant, (ii) the photograph seen, and (iii) its accompanying vignette. The random intercept for vignette was removed for analyses in which the model was unable to converge due to low variance across the control vignettes. All data and a step-by-step guide for our statistical analyses (including data cleaning and assumption checking) are available on the Open Science Framework (https://tinyurl.com/ska2qv9). All analyses were conducted in RStudio Version 1.2.1335 [53].

## Results

### Negative plastic surgery effect

Results showed that plastic surgery stimuli (i.e. those that contained the target plastic surgery vignette) were rated lower on traits pertaining to both warmth and morality, warmth only, morality only, and competence than non-plastic surgery stimuli (i.e. those containing control vignettes). They were also rated lower on desirable UH traits and higher on undesirable HN traits. However, participant scores for justice sensitivity and disgust sensitivity did not moderate these relationships (betas and confidence intervals for these non-significant interactions are accessible using our step-by-step guide on the Open Science Framework; https://tinyurl.com/ska2qv9). Furthermore, there was no difference between plastic surgery and non-plastic surgery stimuli for ratings on undesirable UH or desirable HN traits. Results are summarized in Table 1.

### Person perception and attractiveness

Unattractive stimuli (i.e. those containing an unattractive photograph) were rated lower on warmth and morality, warmth only, morality only, and competence than attractive stimuli (i.e. those containing an attractive photograph). They were also rated lower on desirable UH and desirable HN traits, as well as higher on undesirable UH and undesirable HN traits. Results are summarized in Table 2.

**Table 1. The effect of plastic surgery on person perception (i.e. 'negative plastic surgery effect').**

| Dependent Variable | β | CI 95% | Mean Scores and Standard Deviations on Dependent Variables | |
|---|---|---|---|---|
| | | | Plastic Surgery Stimuli | Non-Plastic Surgery Stimuli |
| Warmth and Morality | -0.12 | [-0.23, -0.004]* | 3.33 (0.79) | 3.43 (0.79) |
| Warmth | -0.11 | [-0.20, -0.02]* | 3.26 (0.78) | 3.36 (0.81) |
| Morality | -0.11 | [-0.16, -0.06]* | 3.38 (0.75) | 3.47 (0.72) |
| Competence | -0.09 | [-0.14, -0.04]* | 3.25 (0.76) | 3.33 (0.75) |
| Desirable UH | -0.08 | [-0.14, -0.02]* | 3.35 (0.77) | 3.42 (0.76) |
| Desirable HN | -0.04 | [-0.13, 0.06] | 3.39 (0.76) | 3.43 (0.81) |
| Undesirable UH | 0.06 | [-0.02, 0.14] | 3.02 (0.89) | 2.96 (0.92) |
| Undesirable HN | 0.24 | [0.10, 0.40]* | 3.34 (0.75) | 3.15 (0.78) |

*Note.* UH = uniquely human, HN = human nature.

*95% confidence intervals for unstandardized regression coefficients that do not include zero.

## Plastic surgery and attractiveness interaction

Except for undesirable HN, there were significant interactions between plastic surgery status and attractiveness on all outcome variables. Specifically, there was evidence for moderation: attractive plastic surgery stimuli were rated less favorably than attractive non-plastic surgery stimuli, but plastic surgery status did not influence person perception for unattractive people (i.e. they were not penalized for planning to have surgery; see Table 3). In other words, attractive women seeking plastic surgery were perceived less favorably than attractive women not seeking plastic surgery, but perceptions for unattractive women remained unchanged regardless of whether they were seeking plastic surgery or planning to complete control activities. Participant scores for justice sensitivity and disgust sensitivity did not moderate any of these interactions (betas and confidence intervals for these non-significant interactions are accessible using our step-by-step guide on the Open Science Framework; https://tinyurl.com/ska2qv9).

## Gender effects

At the request of a reviewer, we also examined whether any of the aforementioned results were moderated by participant gender. We found that only the relationship between plastic surgery

**Table 2. The Effect of attractiveness on person perception.**

| Dependent Variable | β | CI 95% | Mean Scores and Standard Deviations on Dependent Variables | |
|---|---|---|---|---|
| | | | Unattractive Stimuli | Attractive Stimuli |
| Warmth and Morality | -0.19 | [-0.37, -0.01]* | 3.33 (0.82) | 3.48 (0.75) |
| Warmth | -0.47 | [-0.67, -0.29]* | 3.15 (0.83) | 3.53 (0.73) |
| Morality | -0.23 | [-0.36, -0.10]* | 3.37 (0.76) | 3.53 (0.70) |
| Competence | -0.55 | [-0.69, -0.40]* | 3.10 (0.79) | 3.51 (0.65) |
| Desirable UH | -0.20 | [-0.33, -0.06]* | 3.33 (0.80) | 3.48 (0.72) |
| Desirable HN | -0.63 | [-0.81, -0.46]* | 3.17 (0.83) | 3.67 (0.68) |
| Undesirable UH | 0.45 | [0.13, 0.37]* | 3.08 (0.88) | 2.86 (0.93) |
| Undesirable HN | 0.17 | [0.09, 0.25]* | 3.27 (0.74) | 3.13 (0.81) |

*Note.* UH = uniquely human, HN = human nature.

*95% confidence intervals for unstandardized regression coefficients that do not include zero.

**Table 3. The Combined effect of plastic surgery status and attractiveness on person perception.**

| Dependent Variable | β | CI 95% | Mean Differences (Plastic Surgery Condition Minus Non-Plastic Surgery Condition) | |
|---|---|---|---|---|
| | | | Unattractive Stimuli | Attractive Stimuli |
| Warmth and Morality | 0.34 | [0.21, 0.46]* | 0.02 | -0.22^ |
| Warmth | 0.20 | [0.08, 0.32]* | -0.02 | -0.18^ |
| Morality | 0.22 | [0.10, 0.34]* | 0.01 | -0.18^ |
| Competence | 0.20 | [0.09, 0.31]* | 0.01 | -0.17^ |
| Desirable UH | 0.26 | [0.14, 0.38]* | 0.03 | -0.17^ |
| Desirable HN | 0.14 | [0.02, 0.26]* | 0.01 | -0.10^ |
| Undesirable UH | -0.17 | [-0.29, -0.06]* | 0.01 | 0.11^ |
| Undesirable HN | -0.11 | [-0.23, 0.01] | 0.19 | 0.20 |

*Note.* UH = uniquely human, HN = human nature.

* 95% confidence intervals for unstandardized regression coefficients that do not include zero.

^ significant simple effect ($p < .05$); plastic surgery status affects outcome variable rating.

condition and morality but not warmth was moderated by gender, such that the relationship was only significant when participants were women. All other analyses were unaffected by gender and thus it was not included in reported models.

## Discussion

### Hypothesis 1 (primary hypothesis): The negative plastic surgery effect

The present study built on existing literature by examining whether women seeking plastic surgery are systematically perceived differently to other women. We hypothesized that there would be a negative plastic surgery effect; that is, women planning to have plastic surgery would be considered less warm, competent, moral, and human than those planning to complete control activities. Results largely supported this hypothesis. Importantly, this study was the first of its kind in which perceptions of women *planning to undergo* plastic surgery were explored (where focus has previously been on perceptions of women who had *already undergone* plastic surgery). As such, the present study demonstrates that negative attitudes toward plastic surgery extend specifically to plastic surgery itself, and not just to its associated outcomes; that is, negative attitudes toward plastic surgery are not dependent on how women look or feel after surgery, but rather pertain simply to the decision to undergo plastic surgery in the first place. Implications for women seeking plastic surgery are discussed below.

**Low warmth and competence: A recipe for contempt.** The Stereotype Content Model proposes that we form impressions of others by assessing them across two fundamental dimensions: warmth and competence [38, 54]. In this model, individuals are perceived as belonging to one of four quadrants: High Warmth-Low Competence, High Warmth-High Competence, Low Warmth-High Competence, or Low Warmth-Low Competence. Correlational and experimental evidence has demonstrated that the way in which people relate to members of each quadrant is unique [54]. For instance, the Low Warmth-Low Competence quadrant is said to house 'free-loaders' who induce contempt [37, 38]. In line with our hypothesis, plastic surgery stimuli were considered both less warm and less competent than non-plastic surgery stimuli in the present study. As such, we infer that by choosing to undergo plastic surgery, women might be subject to contempt. In work contexts, receiving contemptuous feedback has been associated with decreased self-esteem and increased interpersonal aggressiveness toward colleagues [55]. Likewise, feelings of contempt are one of the primary

predictors of marital breakdown [56, 57]. As such, women seeking plastic surgery (and, by extension, inducing contempt) may face implications both interpersonally and professionally.

**Immorality, dehumanization, and psychological needs.** Results indicated that plastic surgery stimuli were rated lower on morality and humanness traits than non-plastic surgery stimuli, supporting our hypothesis. Literature has suggested that 'feeling moral' ought to be categorized as a basic psychological need. Specifically, Prentice and colleagues [58] found that having a positive perception of one's own morality was uniquely predictive of wellbeing over and above the effect of fulfilling traditional psychological needs (i.e. autonomy, relatedness, and competence). In line with socialization theory (whereby the beliefs of those around us heavily inform our own), we induce that women considering plastic surgery may struggle to 'feel moral' if others perceive them to be morally questionable [59]. In turn, they may experience worsened wellbeing relative to if they were not seeking plastic surgery. Similarly, our results suggest that plastic surgery is dehumanizing. While existing literature has extensively explored the motivations preceding dehumanization, relatively less attention has been paid to the consequences of dehumanization for victims. That said, there is some evidence to suggest that being dehumanized negatively impacts one's ability to meet four psychological needs: perceived control, meaningful existence, sense of belonging, and self-esteem [45]. Further, dehumanized people may experience less empathy from others and be targets for aggression [46]. Therefore, we conclude that dehumanized plastic surgery recipients may face negative psychosocial outcomes.

## Hypothesis 2: Justice sensitivity and disgust sensitivity as moderators for the negative plastic surgery effect

Beyond simply establishing the existence of a negative plastic surgery effect, the present study also sought to explain for whom this effect was strongest. Specifically, we hypothesized that the negative plastic surgery effect would be greater for participants higher in justice sensitivity and disgust sensitivity; that is, those more sensitive to injustice and disgust would perceive women intending to have plastic surgery less favorably. This hypothesis was not supported by results. Firstly, contrary to existing literature, these findings do not support the theory that the negative plastic surgery effect is driven by concerns pertaining to recipients reaping 'unearned' rewards from plastic surgery (i.e. the concern that plastic surgery is 'cheating') [11]. Further, also in contrast with existing literature, these findings do not support the theory that the negative plastic surgery effect is driven by stigmatization toward individuals with non-normative bodies [31, 33, 60, 61]. We note, however, that in the present study, stimuli described as having had plastic surgery did not necessarily *look* non-normative (i.e. they did not look different to control stimuli). As such, we cannot definitively conclude that perceptions of disgust do not drive the negative plastic surgery effect in cases where women have *visibly* undergone surgery (e.g., they look 'artificial'); rather, only in cases where plastic surgery status becomes known via vignette.

## Hypothesis 3: 'What is beautiful is good'

Existing literature demonstrates that the social consequences of being attractive are overwhelmingly positive [22–26]. As such, we hypothesized unattractive stimuli would be perceived as less warm, competent, moral, and human than attractive stimuli. This hypothesis was supported by results. These findings contributes to an extensive and growing body of literature that demonstrates that 'what is beautiful is good'.

### Hypothesis 4: Exploratory hypotheses

**Hypothesis 4a.** Next, we assessed the exploratory hypothesis that the negative plastic surgery effect would be moderated by patient attractiveness; that is, we examined whether attractive and unattractive women planning to have plastic surgery were both subject to similar negative plastic surgery effects. Results indicated that the negative plastic surgery effect applied exclusively to attractive plastic surgery recipients. To elaborate, person perception for unattractive individuals remained unchanged by plastic surgery status (e.g., an unattractive woman planning to have a conversation and an unattractive woman planning to have plastic surgery were perceived similarly), while attractive plastic surgery stimuli were perceived as less warm, competent, moral, and human than attractive non-plastic surgery stimuli.

Because empathy plays a crucial role in reducing stigmatization, we theorize that empathy might explain the revealed interaction between plastic surgery status and stimuli attractiveness [62–67]. Intuitively, individuals may feel that it is more 'understandable' that unattractive women might seek plastic surgery. Given that both the present study's results and existing literature suggest that unattractive women are perceived to be less warm, competent, moral, and human than attractive women, it may seem reasonable for these women to want to undergo plastic surgery to reduce their experiences of appearance-based stigmatization [9, 22]. Conversely, participants may have less empathy for attractive women who do not stand to face the same stigmatization with or without surgery. In line with this theory, existing literature has demonstrated that people feel more empathy toward unattractive individuals (versus attractive individuals) across a variety of situations because they are more easily able to believe that unattractive individuals are suffering or in need of help [66, 67]. As such, we propose that there unattractive stimuli in the present study were not subject to the negative plastic surgery effect because participants were more easily able to empathize with them.

**Hypothesis 4b.** Given that our hypothesis 4a was supported, we subsequently examined whether participant justice sensitivity and/or disgust sensitivity would influence the interaction between plastic surgery status and attractiveness. This exploratory hypothesis, however, was not supported by results. As such, we concluded that neither justice sensitivity nor disgust sensitivity influenced the phenomenon whereby solely attractive individuals were subject to the negative plastic surgery effect.

### Limitations

There were some limitations for the present study. Firstly, we note that the plastic surgery vignette used in our study ("this woman is planning to have plastic surgery") neither specified the nature of the plastic surgery the woman was planning to have, nor the specific surgery performed. We assumed (but did not ensure) that participants would respond to our measures with regard to cosmetic plastic surgery as opposed to reconstructive plastic surgery, given that the faces presented in our stimuli did not look disfigured in any way. We also did not specify whether the plastic surgery in question was for the face (e.g., rhinoplasty) or body (e.g., abdominoplasty), nor provide any other information pertaining to the surgery (e.g., whether she was planning to have one surgery or multiple). As such, the biggest limitation for the present study is that we cannot say with certainty whether the perceptions measured are in relation to cosmetic plastic surgery or reconstructive plastic surgery (or both), and/or whether different specific surgeries would elicit different attitudes from participants (e.g., face vs body).

We also note limitations in the generalizability of our conclusions. The means on all outcome measure scales used in the present study were consistently above the mid-point, regardless of plastic surgery condition, and our effect sizes were consistently small. In other words, while there were statistically significant differences between perceptions of women who seek

surgery and women who do not across all outcome measures, the absolute difference in perceptions of these women were minimal. As such, negative outcomes faced by women seeking plastic surgery may ultimately be small, though still significant and important. Speaking further to the generalizability of the study, we note that only White plastic surgery stimuli were used. These findings therefore cannot be generalized to people of color; specifically, we are unable to establish whether plastic surgery recipients who are people of color are subject to the negative plastic surgery effect. Given that plastic surgery has historically attempted to produce more stereotypically White features (e.g., surgeries for the 'Jewish nose' or 'Black nose'), it is especially important that we acknowledge the limited applicability of our findings [32, 68].

## Implications, conclusions, and future directions

The present study demonstrates the existence of a negative plastic surgery effect, specifically for attractive women. In planning to undergo plastic surgery, these women are perceived as less warm, moral, competent, and human. As such, we contend that attractive women seeking plastic surgery may find themselves experiencing negative psychosocial outcomes (e.g., being subject to contempt). However, we note that at present these outcomes are purely speculative, and that future research is needed to test these associations. As per our limitations section, future research also ought to examine the negative plastic surgery for cosmetic plastic surgeries and reconstructive plastic surgeries separately, and for different types of surgeries (e.g., face vs body). Future research might also explore additional consequences that women subject to the negative plastic surgery effect are likely to face. For example, might this worsened person perception result in social exclusion or prejudicial treatment? Finally, future research need address whether these results are generalizable to non-White plastic surgery recipients. Overall, our study was the first to examine the negative plastic surgery effect experimentally. We provide a fundamental starting point from which future literature can further investigate negative plastic surgery attitudes in order to inform both women seeking plastic surgery and plastic surgeons themselves.

## Acknowledgments

The authors would like to thank Christoph Klebl for his assistance launching the study.

## Author Contributions

**Conceptualization:** Sarah Bonell, Scott Griffiths.

**Data curation:** Sean C. Murphy.

**Formal analysis:** Sarah Bonell, Sean C. Murphy.

**Investigation:** Sarah Bonell.

**Methodology:** Sarah Bonell, Sean C. Murphy, Scott Griffiths.

**Supervision:** Scott Griffiths.

**Writing – original draft:** Sarah Bonell.

**Writing – review & editing:** Sarah Bonell, Sean C. Murphy, Scott Griffiths.

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
