## [Decision Letter · Decision Letter 0]

15 Apr 2021

PONE-D-21-05038

Unfavorable Perceptions of Women Who Seek Plastic Surgery

PLOS ONE

Dear Dr. Bonell,

Thank you for submitting your manuscript to PLOS ONE. After careful consideration, we feel that it has merit but does not fully meet PLOS ONE’s publication criteria as it currently stands. Therefore, we invite you to submit a revised version of the manuscript that addresses the points raised during the review process.

We look forward to receiving your revised manuscript.

Kind regards,

Ali B. Mahmoud, Ph.D.

Academic Editor

PLOS ONE

Journal Requirements:

Reviewers' comments:

Reviewer's Responses to Questions

**Comments to the Author**

1. Is the manuscript technically sound, and do the data support the conclusions?

Reviewer #1: Yes

Reviewer #2: Yes

Reviewer #3: Partly

2. Has the statistical analysis been performed appropriately and rigorously? 

Reviewer #1: No

Reviewer #2: Yes

Reviewer #3: Yes

3. Have the authors made all data underlying the findings in their manuscript fully available?

Reviewer #1: Yes

Reviewer #2: Yes

Reviewer #3: Yes

4. Is the manuscript presented in an intelligible fashion and written in standard English?

Reviewer #1: Yes

Reviewer #2: Yes

Reviewer #3: Yes

5. Review Comments to the Author

Reviewer #1: Unfavorable Perceptions of Women Who Seek Plastic Surgery

The manuscript presents an original article exploring how women seeking plastic surgery are perceived by others, which is certainly of interest to readers of PLOS ONE, since it is according the journal scope. Although it was highlight, that perceptions of attractive women are worsened when these women decide to seek cosmetic surgery, authors only refer that psychoeducation content can be developed for women considering plastic surgery, without specifying which ones. I have several comments and suggestions that I think, when addressed, would improve the manuscript. These are outlined below.

Title:

Should be written in sentence case (only the first word of the text, proper nouns, and genus names are capitalized).

Abstract:

. Line 15: “….985 adults…” – what kind of sample (random, convenience)? Please explain.

. Line 16: “…M age…” Please add the standard deviation.

. Line 19: “…humanness”. In the end of the sentence, please refer the statistical analyses performed.

. Line 21: “… control activities …” Please specify/give examples.

Introduction:

Despite being presented, in theoretical terms, the contextualization of the problem, it is not elaborated, in practical terms, what this investigation can contribute to the resolution of the problem. Please explain to what extent the perception of others about plastic surgery is important and how it may have practical implications for those who intend to perform plastic surgery.

. Line 67: “…Brazilian women”. Please cite reference.

. Line 84 – “…psychoeducation content…”: Please elaborate / specify.

. Lines 173-175; 177-178: Very confusing phrases: after all you intend to evaluate the perception of women who intend to have a plastic surgery or the perception of others about women who intend to have plastic surgery? Please clarify since it is incongruous with the presented hypotheses.

Since the participants were from the two sex, It would be interesting to explore if women seeking plastic surgery are perceived by males and females in the same way.

Material and Methods:

. Line 196: Explain how the sample size was calculated. If a sample size calculation was performed, specify the inputs for power, effect size and alpha.

. Lines 202-203: ”… see …data quality). This sentence should be cut and only the reference should appear. In limitation you should refer this.

. Lines 206-207: ”… see …checks). This sentence should be cut and only the reference should appear. Justify in the text, why you excluded 16 participants and cite the reference.

. Line 227: How many participants for the pilot study?

. Line 298: Statistical Analysis:

. What is the assumptions criterion used for LMM?

. Describe any analysis carried out to confirm the data meets the assumptions of the analysis performed (e.g.: linearity, co-linearity, normality of the distribution).

. Please explain in detail how was made the data generation (Describe the technical details or procedures required to reproduce the analysis)

. Please indicate what was the statistical package software used to carry out LMM analyses (List the name and version of software package used, alongside any relevant references).

Results:

. Lines 313-314: “...these interactions). Please indicate the ß and CI 95%.

. Line 326: “…were also were…” cut the second “were”.

. Lines 340-341: “...these relationships). Please indicate the ß and CI 95%.

Discussion:

. Lines 349-350: Again, very confusing phrase: you intend to evaluate the perception of women who intend to have a plastic surgery or the perception of others about women who intend to have plastic surgery? Please clarify

. Line 353: “…control activities”: Please give examples.

. Line 369: “…is unique” Please cite references.

. Lines 376-377: “…seeking plastic surgery is likely to foster implication for women …”. I wonder, although this is a possibility in the face of the choice of a woman to decide to have plastic surgery, something that in principle should only concern her, the fact that she does not do it for the sake of others, will have milder or more serious psychosocial implications/impact for the own? Please elaborate about it.

. Line 385: “…inform our own”. Please cite reference.

. Line 387: “…wellbeing relative to …”. Relative to what?? Please explain.

. Line 394: “…and that this in turn …”. Cute “that this”.

. Lines 400-401: “…was not …by results”. Why? What is the possible explanation? Please elaborate.

. Lines 417-419: “As such, …surgery effect”. Why? What is the possible explanation? Please elaborate.

. Lines 426, 435, 449: “To elaborate”. Please avoid the systematic repetition of the expression.

. Lines 460-463: The author refers to women perception who intend to have a plastic surgery or the perception of others about women who intend to have plastic surgery? Please clarify!

. Limitation: As you refer on the participants section, it was made a compensation for each participant completing the survey. Taking this in account, how do you guarantee honest responses?

. Line 471: “…psychoeducation content…”. Please elaborate / specify.

. Lines 473-475: “… we suspect…planning to do so”. I am confused… taking in account that this study “… posits that exploring plastic surgery as an intended action would enable researchers to better assess negative perceptions …rather than its associated outcomes”, how do you explain your suspicious??

Lines 478-479: “Overall…experimentally”. By whom?? Please refer.

Lines 479-480: “we…attitudes”. Please highlight the importance of the thematic, concerning future studies.

References/Citations:

. Text: If the intention is to use Vancouver style consistently, please revise its correct use and format throughout the text. Special attention should be given to citations of references (e.g., Lines: 56, 74, 100, 124, 135, 151, 163, 203 (this one is in APA style), 207, 215…).

Please revise “DOI” number and adhere to the format give in “PLOS ONE Submission Guidelines”.

Tables:

The titles should not be in italic and should be align with table identification.

Reviewer #2: The paper is well presented and documented. The authors show a good grasp of the theoretical context and the data presented is solid and persuasive. The research design is good and the conclusions are generally sound.

Reviewer #3: The premise of this research and the methods used (i.e. rating women on attractiveness) is steeped in misogyny and has no place in contemporary scholarship. We must do better.

See this media piece for a summary of the types of issues that are promoted by your work: https://www.theguardian.com/society/2020/aug/05/disgusting-study-rating-attractiveness-of-women-with-endometriosis-retracted-by-medical-journal

6. PLOS authors have the option to publish the peer review history of their article (what does this mean?). If published, this will include your full peer review and any attached files.

Reviewer #1: No

Reviewer #2: No

Reviewer #3: No

---

## [Author Response · Author response to Decision Letter 0]

28 Apr 2021

Reviewer 1

R1.1 Miscellaneous

Although it was highlight, that perceptions of attractive women are worsened when these women decide to seek cosmetic surgery, authors only refer that psychoeducation content can be developed for women considering plastic surgery, without specifying which ones.

We have made clearer throughout our manuscript the practical implications that may arise from our findings. Namely, we have shifted focus away from the generation of psychoeducation content. Instead, we focus on how the present findings might shape our understanding of plastic surgery attitudes, and how women who might be considering plastic surgery can benefit from our findings. We feel these implications are more realistic and more in line with the aims of the present study. See changes throughout our manuscript and see below for an example of these changes.

e.g., “This study is one of few that elucidates the potential psychosocial challenges women seeking plastic surgery might face. As such, findings from this study might prove informative over and above any information women might receive from surgeons or surgical clinics (i.e. where the focus in on physical outcomes only) and/or might work to educate surgeons themselves so that they are better able to inform their clientele. Specifically, we feel that from an ethical standpoint it is important for women seeking cosmetic surgery to fully understand the implications of their decisions (e.g., in undergoing cosmetic surgery, they may be perceived as less human, among other things).” p25

R1.2 Title

Should be written in sentence case (only the first word of the text, proper nouns, and genus names are capitalized).

We applied this revision. 

“Under the knife: Unfavorable perceptions of women who seek plastic surgery” p1

R1.3 Abstract

R1.3.1 Line 15: “….985 adults…” – what kind of sample (random, convenience)? Please explain.

We have specified that our sample was random. 

“We presented a random sample of 985 adults (male = 54%, Mage = 35.84 years) recruited via Amazon’s Mechanical Turk with a series of experimental stimuli” p2

R1.3.2 Line 16: “…M age…” Please add the standard deviation.

We have added standard deviation 

“(male = 54%, Mage = 35.84 years, SDage = 10.59)” p2

R1.3.3 Line 19: “…humanness”. In the end of the sentence, please refer the statistical analyses performed.

We have included our analysis. 

“Participants rated stimuli on perceived warmth, competence, morality, and humanness. We ran linear mixed-effect models to assess all study hypotheses.”p2

R1.3.4 Line 21: “… control activities …” Please specify/give examples.

We feel that adding more information pertaining to the control activities in the abstract would confuse readers more so than aid clarity. However, we have added an additional example of for control activities in the Materials and Measures section of our Method. An exhaustive list of our control vignettes has now also been uploaded to the OSF, and this has been indicated in our manuscript. 

“Of these, 12 were control vignettes that depicted a woman planning to undergo a neutral – that is, common or everyday – activity (e.g., “this woman is planning to eat a meal”, “this woman is planning to buy a pet”). The remaining vignette was our plastic surgery target vignette – “this woman is planning to have plastic surgery”. Again, each participant would read four of these vignettes in total; three control vignettes (of the 12 in total) and the plastic surgery vignette. Further details are outlined in the procedure. A full list of control vignettes can be found on the Open Science Framework (https://tinyurl.com/ska2qv9).” p11

R1.4 Introduction

R.1.4.1 Despite being presented, in theoretical terms, the contextualization of the problem, it is not elaborated, in practical terms, what this investigation can contribute to the resolution of the problem. Please explain to what extent the perception of others about plastic surgery is important and how it may have practical implications for those who intend to perform plastic surgery.

See our response to R1.1 for an overview of how we included more practical implications in our discussion. For our introduction, we also included this line:

“This knowledge will help contribute to our growing understanding of how society perceives plastic surgery, thereby elucidating the challenges women who choose to undergo it are likely to face. Without this study, uninformed women may choose to seek out plastic surgery and find themselves surprised and disappointed by the negative psychosocial consequences they face postoperatively.” p5

R1.4.2 Line 67: “…Brazilian women”. Please cite reference.

We have added three references.

“…power for Brazilian women; 13–15).” p4

R1.4.3 Line 84 – “…psychoeducation content…”: Please elaborate / specify.

This line has been removed from the manuscript, rendering this suggestion obsolete. See R1.1 for removal justification. 

R.1.3.4 Lines 173-175; 177-178: Very confusing phrases: after all you intend to evaluate the perception of women who intend to have a plastic surgery or the perception of others about women who intend to have plastic surgery? Please clarify since it is incongruous with the presented hypotheses.

We have reworded these lines. 

“In an attempt to examine the social consequences of undergoing plastic surgery, the present paper will therefore examine how women who seek plastic surgery are perceived across four domains: warmth, competence, morality, and humanness … We aimed to examine whether perceptions of women who seek plastic surgery systematically differ from perceptions of women who do not.”p9

R1.3.5 Since the participants were from the two sex, it would be interesting to explore if women seeking plastic surgery are perceived by males and females in the same way.

We examined the effect of participant gender on the negative plastic surgery effect. While effects were non-significant for most outcome variables (warmth, competence, and humanness), we did find that plastic surgery status only predicted morality scores in cases where participants were female. In other words, women seeking plastic surgery were only considered less moral than those not seeking surgery by female participants.

While ultimately we did not decide to include this exploratory analysis in our revised manuscript (given that it largely produced non-significant results and was not part of our initial analysis plan), we do value improving our understanding of these variables and their interactions and thank the reviewer for their suggestion. 

R1.5 Material and Methods

R1.5.1 Line 196: Explain how the sample size was calculated. If a sample size calculation was performed, specify the inputs for power, effect size and alpha.

No sample size calculation was performed for this study. In hindsight, we understand that ideally a sample size calculation would have been performed. However, in the interest of transparency, we note that sample size was informed exclusively by funding available to our lab at the time of data collection. We have indicated this in the manuscript. 

“Sample size was determined prior to data analysis based on funding available to our research team at the time of data collection.”p10

R1.5.2 Lines 202-203: ”… see …data quality). This sentence should be cut and only the reference should appear. In limitation you should refer this.

We have amended this sentence. We do not believe our compensation to be a limitation; rather, we were attempting to refer readers to this article in case they were wondering why we compensated participants more generously than most other studies on MTurk. 

“Compensation for each participant completing our 15-minute survey was US$2.33 (42).”p10

R1.5.3 Lines 206-207: ”… see …checks). This sentence should be cut and only the reference should appear. Justify in the text, why you excluded 16 participants and cite the reference.

We have amended this. 

“We excluded 16 participants from the present study for failing to consistently report their race. To elaborate, we included two items in our survey asking participants to indicate their race – one toward the commencement, and one toward the conclusion, of the survey. In cases where participant responses did not align between these two items, ‘participants’ were assumed to be bots (or to simply not be paying attention) and were subsequently removed from the study (43).” p10

R1.5.4 Line 227: How many participants for the pilot study?

208. We have included this in the manuscript. This information is also available in the data file on the Open Science Framework. 

“All vignettes were assessed for their ability to induce participant affect and arousal in an MTurk pilot study (N = 208)” p11

R1.5.5 What is the assumptions criterion used for LMM?

Before interpreting the output of any multilevel models, we checked the key assumptions of linearity of relationships (where relevant, as many of our analyses had binary predictors only, thus linearity was inherent), normality of residuals, and homoscedasticity. For these checks, we took a graphical approach as recommended by Fife (2020). We used the R-packagesjPlot (Lüdecke,2020) to draw diagnostic plots. To check for linearity, we plotted model residuals against predictors. To check for normality in fixed effects, we plotted a Q-Q plot of residuals and the distribution of residuals. To check for normality in random effects, we plotted random effect quantiles against standard normal quantiles. To check for homoscedasticity, we plotted fitted values against residuals. All assumptions fell within reasonable bounds upon visual inspection.

R1.5.6 Describe any analysis carried out to confirm the data meets the assumptions of the analysis performed (e.g.: linearity, co-linearity, normality of the distribution).

See R1.5.5

R1.5.7 Please explain in detail how was made the data generation (Describe the technical details or procedures required to reproduce the analysis)

This is available on the Open Science Framework. We had noted this in our manuscript, but have also amended the wording to make it clear that a full guide is provided on the Open Science Framework. 

“All data and a step-by-step guide for our statistical analyses (including data cleaning) are available on the Open Science Framework (https://tinyurl.com/ska2qv9).”p15

R1.5.8 Please indicate what was the statistical package software used to carry out LMM analyses (List the name and version of software package used, alongside any relevant references).

RStudio Version 1.2.1335 was used.

“All analyses were conducted in RStudio Version 1.2.1335 (48).”p15

R1.6 Results

R1.6.1 Lines 313-314: “...these interactions). Please indicate the ß and CI 95%.

ß and CI 95% can be calculated using the step-by-step guide we provide on the Open Science Framework. We feel as though providing all ßs and CIs in our manuscript would add unnecessary confusion and bulk to our text. However, we have added a note in our manuscript that these statistics are accessible using the Open Science Framework. 

“Participant scores for justice sensitivity and disgust sensitivity did not moderate any of these interactions. Betas and confidence intervals for these non-significant interactions are accessible using our step-by-step guide on the Open Science Framework (https://tinyurl.com/ska2qv9).” p15

R1.6.2 Line 326: “…were also were…” cut the second “were”.

We’ve cut out this typo. 

R1.6.3 Lines 340-341: “...these relationships). Please indicate the ß and CI 95%.

See R1.6.1 

“However, participant scores for justice sensitivity and disgust sensitivity did not moderate these relationships. Betas and confidence intervals for these non-significant interactions are accessible using our step-by-step guide on the Open Science Framework (https://tinyurl.com/ska2qv9).” p18

R1.7 Discussion

R1.7.1 Lines 349-350: Again, very confusing phrase: you intend to evaluate the perception of women who intend to have a plastic surgery or the perception of others about women who intend to have plastic surgery? Please clarify

Again, we have amended our wording.

“The present study built on existing literature by examining whether women seeking plastic surgery are systematically perceived differently to other women.” p19

R1.7.2 Line 353: “…control activities”: Please give examples.

Given that we have reiterated that control activities can be found on the Open Science Framework, as well as included more examples in our method section (see R1.3.4), we don’t feel it necessary to here again include examples of control activities. However, if the reviewer ultimately disagrees, we will happily amend to include the discussion to include examples. 

R1.7.3 Line 369: “…is unique” Please cite references.

We have included a reference for this sentence.

“…has demonstrated that the way in which people relate to members of each quadrant is unique (37).” p19

R1.7.4 Lines 376-377: “…seeking plastic surgery is likely to foster implication for women …”. I wonder, although this is a possibility in the face of the choice of a woman to decide to have plastic surgery, something that in principle should only concern her, the fact that she does not do it for the sake of others, will have milder or more serious psychosocial implications/impact for the own? Please elaborate about it.

We agree that it would be interesting to investigate whether women’s motivations for undergoing surgery (i.e. for themselves versus for others) would affect how the negative plastic surgery effect might impact their psychosocial wellbeing. However, because this area of research is so underexamined, we feel there are an abundance of potential, unexplored variables that might moderate this relationship (e.g., surgical outcome, recipient gender, recipient age etc). As such, we feel it is outside the scope of our paper to address all of these moderators, and would prefer not to incorporate one in lieu of others. 

R1.7.5 Line 385: “…inform our own”. Please cite reference.

This line is already cited – perhaps this was an accidental oversight by the reviewer.

“In line with socialization theory (whereby the beliefs of those around us heavily inform our own), we induce that women considering plastic surgery may struggle to ‘feel moral’ if others perceive them to be morally questionable (56).” p20

R1.7.6 Line 387: “…wellbeing relative to …”. Relative to what?? Please explain.

Again, we believe this might be a reviewer oversight, as we explain this is relative to if they were not seeking plastic surgery. 

“In turn, they may experience worsened wellbeing relative to if they were not seeking plastic surgery.” p20

R1.7.7 Line 394: “…and that this in turn …”. Cut “that this”.

We believe the original wording of the sentence to be grammatically correct, but added a comma for clarity.

“Overall, we conclude that the ability to fulfil psychological needs may be affected for women seeking plastic surgery, and that this in turn may detriment their wellbeing.” p20

R1.7.8 Lines 400-401: “…was not …by results”. Why? What is the possible explanation? Please elaborate.

We have expanded on this section. See R2.5.

R1.7.9 Lines 417-419: “As such, …surgery effect”. Why? What is the possible explanation? Please elaborate.

Ultimately, given that this was a purely exploratory hypothesis grounded in little existing literature or theory, we have decided not to expand on our interpretation of this result. However, we are happy to make further changes and expand on this section if the reviewer is not happy with our decision. 

R1.7.10 Lines 426, 435, 449: “To elaborate”. Please avoid the systematic repetition of the expression.

We removed one of these sentences from our manuscript, and reworded another to avoid repetition.

R1.7.11 Lines 460-463: The author refers to women perception who intend to have a plastic surgery or the perception of others about women who intend to have plastic surgery? Please clarify!

We have reworded this sentence to aid clarity. 

“In other words, while there were statistically significant differences between perceptions of women who seek surgery and women who do not across all outcome measures, the absolute difference in perceptions of these women were minimal.” p24

R1.7.12 Limitation: As you refer on the participants section, it was made a compensation for each participant completing the survey. Taking this in account, how do you guarantee honest responses?

Providing compensation to participants is generally considered best practice, so long as payment amount is appropriate (i.e. not so large that it is deemed coercive or undue; e.g., see Ripley [2006] for a review). While we cannot guarantee honest responses, we did include an attention check measure in our study to ensure ‘bots’ and distracted workers’ responses were not included in our analyses (see R1.5.3). 

R1.7.13 Line 471: “…psychoeducation content…”. Please elaborate / specify.

We have removed this line. See R1.1.

R1.7.14 Lines 473-475: “… we suspect…planning to do so”. I am confused… taking in account that this study “… posits that exploring plastic surgery as an intended action would enable researchers to better assess negative perceptions …rather than its associated outcomes”, how do you explain your suspicious??

We agree with the reviewer’s concerns regarding this section, and have subsequently removed it from our discussion.

R1.7.15 Lines 478-479: “Overall…experimentally”. By whom?? Please refer.

We suspect the reviewer has misread this sentence in our manuscript, as we are referring to our own study at this point (see below). However, we are happy to revisit the comment if we have misunderstood. 

“Overall, our study was the first to examine the negative plastic surgery effect experimentally.” p25

R1.7.16 Lines 479-480: “we…attitudes”. Please highlight the importance of the thematic, concerning future studies.

We weren’t too sure what the reviewer meant by ‘thematic’, but we have attempted to address this feedback by including more information about future implications.

“We provide a fundamental starting point from which future literature can further investigate negative plastic surgery attitudes in order to inform both women seeking cosmetic surgery and cosmetic surgeons themselves.” p25

R1.8 References/Citations

If the intention is to use Vancouver style consistently, please revise its correct use and format throughout the text. Special attention should be given to citations of references (e.g., Lines: 56, 74, 100, 124, 135, 151, 163, 203 (this one is in APA style), 207, 215…).

Please revise “DOI” number and adhere to the format give in “PLOS ONE Submission Guidelines”.

We have removed page numbers (e.g., line 56). For lines 74, 100, 124, 135, 151, 163, and 215, bracketing has been corrected. We have amended lines 203 and 207. Further, as per PLoS ONE guidelines, DOI numbers can be used in place of page numbers but are not necessary for inclusion if page numbers are already included. As such, we have not added DOIs to our references. 

R1.9 Tables

The titles should not be in italic and should be align with table identification.

We have amended this formatting. 

Reviewer 2

R2.1 Abstract and Introduction: Here I suggest that the authors provide some definition of the terms ‘attractiveness’ and ‘unattractiveness’. If these are major construct unpinning the perceptions that they are seeking to explain, then they should describe what they mean by these terms. It may be that they see attractiveness and unattractiveness from the participants’ subjective appreciation/judgement, but then the authors have to explain that.

We have added a brief note in our introduction to make clear what we mean by attractiveness. Also, we have reiterated how attractiveness was conceptualised in the present study in the materials and measures section of our method. 

“An individual’s physical attractiveness – by which we mean their average attractiveness score as rated by others – affects how others perceive them.” p5

“We compiled a smaller database containing the eight most and eight least attractive (as rated by the 1087 CFD participants) White women contained in the CFD for use in the present study (i.e. 16 photographs in total).” p11

R2.2 Introduction (p.3, line 36). The authors claim that plastic surgery demand and operations have increased significantly. It will be helpful if they provide some data from the literature or health to substantiate this. Also, they could define a broad time period over which such increase can be observed.

We agree, and have made the necessary revisions. 

“Today, approximately 18 million plastic surgeries are performed in the US each year – a 169% increase since the year 2000 (2,3). Further, rates increased by more than 20% worldwide between the years 2015 and 2019 (4).” p3

R2.3 Hypotheses (p.9). Overall, it would be helpful if the authors number their hypotheses for ease of reference from a reader perspective but also for themselves to recapture them in the discussion or conclusion. The hypotheses on the same p.9 (line…) is not clearly related to the research question and the aim of the study which to examine the perception of women seeking plastic surgery or having had plastic surgery. I do see the connection. Perhaps the authors could explain and establish the link clearly.

We have numbered our hypotheses (1, 2, 3, 4a, 4b) for ease of reference. We were not entirely sure whether the editor’s comments pertaining to ‘the hypotheses on the same p.9 (line…)’ referred to a specific hypothesis on p9, or rather our hypotheses in general. However, we posit that all our hypotheses bar hypothesis 3 are directly related to examining the perception of women seeking plastic surgery. Therefore, we assume the reviewer is referring to hypothesis 3. In light of this, we contend that hypothesis 3, while not directly related to our manuscript’s research question, works toward validating existing literature (as stated in the manuscript). Therefore, we feel its inclusion is justified. We hope this response has satisfied the reviewer’s concerns and are happy to make future changes to the manuscript if necessary. 

“Our primary hypothesis was that (1) women seeking plastic surgery would be considered less warm, competent, moral, and human than those who are not (i.e. there would be a negative plastic surgery effect). We also hypothesized that (2) this relationship… (3)… (4a)… (4b)”p9

R2.4 Methods (p.10). The word [in] might be missing before [total] in the following phrase (i.e. 16 photographs total). Same for line 224 on p.11 in the phrase (of the 16 total) and line 270 on p.13 in the phrase (16 items total). Please check these parentheses.

We have applied these revisions to all noted parentheses. 

R2.5 Results. On p.20, the coverage of Justice sensitivity needs elaboration drawing of the data and contrasting with the literature. 

We have elaborated in this section.

“Firstly, contrary to existing literature, these findings do not support the theory that the negative plastic surgery effect is driven by concerns pertaining to recipients reaping ‘unearned’ rewards from plastic surgery (i.e. the concern that plastic surgery is 'cheating'; 11,25,55). Further, also in contrast with existing literature, these findings also do not support the theory that the negative plastic surgery effect is driven by stigmatization toward individuals with non-normative bodies (22,26,27). We note, however, that because of our study design, stimuli described as having had plastic surgery in the present study did not necessarily look non-normative (i.e. they did not look different to control stimuli). As such, we cannot definitively conclude that perceptions of disgust do not drive the negative plastic surgery effect in cases where women have visibly undergone surgery (e.g., they look ‘artificial’); rather, only in cases where plastic surgery status alternatively becomes known.” p21

R2.6 Implications. The authors could elaborate on the practical implications of the study under the heading on p.20. 

See R1.1. We have incorporated our implications into a section titled “Implications, Conclusions, and Future Directions” on page 24. 

R2.7 Future research. I think it would interesting for future research to also look at the consequences of such negative perception of ‘attractive’ women undergoing plastic surgery. Does it lead to physical aggression? Stereotyping? Some forms of exclusion? And what those women affect could do?

We have incorporated this suggestion in the final section of our manuscript. 

“Future research might also explore additional consequences that women subject to the negative plastic surgery effect are likely to face. For example, might this worsened person perception result in the social exclusion or prejudicial treatment?” p24

Reviewer 3

The premise of this research and the methods used (i.e. rating women on attractiveness) is steeped in misogyny and has no place in contemporary scholarship. We must do better. See this media piece for a summary of the types of issues that are promoted by your work: https://www.theguardian.com/society/2020/aug/05/disgusting-study-rating-attractiveness-of-women-with-endometriosis-retracted-by-medical-journal

We appreciate and share reviewer 3’s disdain for the ongoing misogyny rife in contemporary scholarship. Further, we are familiar with the endometriosis study provided as an example by reviewer 3 and also believe this work to be fundamentally problematic. However, we disagree with reviewer 3 with regards to the role our own manuscript plays in promoting misogyny in contemporary research. Notably, while we (assumedly) agree with reviewer 3 in that women should not be judged based on their attractiveness, we feel it is important to acknowledge that they nevertheless are (see Dion’s 1972 seminal “What is beautiful is good”; see commentary on ‘lookism’ by Australian scholar Beatrice Alba – https://tinyurl.com/4fd4azbp). The intent of our manuscript is not to encourage the objectification of women, but rather (in part) to investigate the very real and important role attractiveness plays in person perception. 

Dissimilarly to the endometriosis study referenced by reviewer 3, we feel our manuscript has clear scientific merit. For example, our study works to validate existing literature as well as anecdotal accounts of women’s lived experiences of discrimination based on ‘lookism’ (see commentary above). Further, our study importantly examines whether women seeking cosmetic surgery – potentially to avoid facing discrimination based on their attractiveness – might actually be subject to further discrimination for choosing to undergo surgery. Finally, we note that at no stage during our study were participants actually asked to rate each other (or even photographed stimuli) on attractiveness. Rather, our methodology relied on a set of pre-existing, validated faces already rated on attractiveness (the Chicago Faces Database; CFD). Importantly, CFD participants provided full informed consent to be rated on attractiveness. 

Lastly, we would like to make known to reviewer 3 that all collaborators on this paper are members of what we consider to be a feminist research team. While we may not agree with reviewer 3’s concerns regarding our work, we very much do appreciate the opportunity to engage in feminist dialogue and recognise that differences in opinion will exist even within the feminist community. In an attempt to placate some of reviewer 3’s concerns, we have added a brief note in our introduction to make clear what we mean by ‘attractiveness’ in this context (thereby eliminating any misconceptions that our research team believes beauty to be an entirely objective or dichotomous construct). Further, we have reiterated how attractiveness was conceptualised in the present study in the materials and measures section of our method. We hope that this response has addressed reviewer 3’s concerns; however, we are of course happy to take on board any further suggestions for improving our manuscript. 

“An individual’s physical attractiveness – by which we mean their average attractiveness score as rated by others – affects how others perceive them.” p5

“We compiled a smaller database containing the eight most and eight least attractive (as rated by the 1087 CFD participants) White women contained in the CFD for use in the present study (i.e. 16 photographs in total).” p11

---

## [Decision Letter · Decision Letter 1]

18 Jun 2021

PONE-D-21-05038R1

Under the Knife: Unfavorable Perceptions of Women Who Seek Plastic Surgery

PLOS ONE

Dear Dr. Bonell,

Thank you for submitting your manuscript to PLOS ONE. After careful consideration, we feel that it has merit but does not fully meet PLOS ONE’s publication criteria as it currently stands. Therefore, we invite you to submit a revised version of the manuscript that addresses the points raised during the review process.

Certainly, in the previous round of the peer review, one reviewer raised grave concerns about misogynistic views and recommended rejection because of that. Further, they declined my invitation to review your revision. However, aiming for an ethical decision that would be as fair as possible for everyone engaged in this research and the review process, I invited additional reviewers who are experts in feminist studies and the psychological perspectives of plastic surgeries. In my invitation letters, as well as separate emails, I've asked the new (and the original) reviewers to highlight any 'misogynistic' views that the study might express (I had the report and concerns of the original reviewer who raised this issue shared with all of the reviewers). Based on the comments and recommendations of the current reviewers, whom I thank faithfully, I was able to make a well-informed decision on your paper. 

We look forward to receiving your revised manuscript.

Kind regards,

Ali B. Mahmoud, Ph.D.

Academic Editor

PLOS ONE

Reviewers' comments:

Reviewer's Responses to Questions

**Comments to the Author**

1. If the authors have adequately addressed your comments raised in a previous round of review and you feel that this manuscript is now acceptable for publication, you may indicate that here to bypass the “Comments to the Author” section, enter your conflict of interest statement in the “Confidential to Editor” section, and submit your "Accept" recommendation.

Reviewer #1: (No Response)

Reviewer #2: (No Response)

Reviewer #4: (No Response)

Reviewer #5: (No Response)

Reviewer #6: (No Response)

2. Is the manuscript technically sound, and do the data support the conclusions?

Reviewer #1: Yes

Reviewer #2: Yes

Reviewer #4: Partly

Reviewer #5: Yes

Reviewer #6: Partly

3. Has the statistical analysis been performed appropriately and rigorously? 

Reviewer #1: Yes

Reviewer #2: Yes

Reviewer #4: Yes

Reviewer #5: I Don't Know

Reviewer #6: Yes

4. Have the authors made all data underlying the findings in their manuscript fully available?

Reviewer #1: Yes

Reviewer #2: Yes

Reviewer #4: Yes

Reviewer #5: Yes

Reviewer #6: Yes

5. Is the manuscript presented in an intelligible fashion and written in standard English?

Reviewer #1: Yes

Reviewer #2: Yes

Reviewer #4: Yes

Reviewer #5: Yes

Reviewer #6: Yes

6. Review Comments to the Author

Reviewer #1: Overall, the authors have successfully addressed my previous concerns and comments.

However, there are still some points that need clarification or revision:

- Pg 12, Line 258 -259: Please put in () the sentence “ A full list…”.

- Pg 16, Statistical Analyses: Include in the text, the assumptions criterion used for LMM, according to your reviewer answer.

- Pg 16, Line 350 -352: Please put in () the sentence “ Betas…ska2..”.

- Pg 19, Line 381 -383: Please put in () the sentence “ Betas…ska2..”.

- Pg 24, Line 462 -463: I appreciate that authors expand their interpretation concerning the obtained result and give examples with references.

- Pg 28, Line 558 -561: Please cut the sentence “ Specifically…things”, as it is sufficient what was already said above.

Reviewer #2: The authors needs to attend to some minor comments as highlighted in attached reviewer comment sheet.

Reviewer #4: Review:

Thank you for the opportunity to review the present manuscript: Under the knife: Unfavorable perceptions of women who seek plastic surgery. The manuscript reports on an online experimental study examining perceptions of attractive and unattractive women planning to engage in plastic surgery or control activities. The project is interesting and has sufficient depth, and I applaud the authors’ transparency and engagement with open practices, as well as their considerate and deep engagement with the prior reviews. I believe the manuscript is fundamentally sound, and I do not believe the present manuscript expresses or reifies misogynist views.

However, I do believe the work could benefit from consideration of the following points. I indicate one primary theoretical concern regarding how the implications, as presently framed, flow from the findings. I believe reevaluation of these aspects of the manuscript will assist in tackling any concerns regarding the perspectives put forth in the manuscript. This primary concern is followed by miscellaneous smaller concerns and suggestions. I hope the authors find these suggestions useful for better presenting their research.

My primary concern with the work pertains to the framing of the implications. I think the authors could grapple more with a few specific issues and I believe that doing so would clarify the feminist intentions of the work. As it is presently written, the paper does seem to suggest that women are responsible for the perceptions/stigmatization from others. In particular, that the practical implications of the work suggest intervention for women seeking plastic surgery (e.g., Without this study, uninformed women may choose to seek out plastic surgery and find themselves surprised and disappointed by the negative psychosocial consequences they face postoperatively) rather than intervention for prejudiced audiences. In my reading, this places the burden on the stigmatized group to change, rather than vying for structural change. The authors are not incorrect to suggest that work of the present nature may help to inform women of potential negative outcomes, however, I think the current framing of these implications is slightly inappropriate. To provide a parallel example: it is broadly understood that fat people face significant stigma, however we do not suggest that we should educate people not to get fat so that they will not have to endure that stigma. Broadly, the work would benefit from more in-depth feminist theorization regarding relevant processes of stigma and prejudice.

It may be relevant in the introduction to provide plastic surgery statistics specifically for women – men make up an increasing amount of plastic surgery patients, which is important to acknowledge, but also suggests that the statistics presented in the introduction are not particularly relevant to the current study’s focus on women and plastic surgery.

p.5 “Without this study, uninformed women may choose to seek out plastic surgery and find themselves surprised and disappointed by the negative psychosocial consequences they face postoperatively”. – this is a very strong claim. First, the authors suggest that stigma surrounding plastic surgery is widespread; to make the present claim, the authors should demonstrate that people are not generally aware of this plastic surgery stigma. Second, the authors seem to be suggesting (a) that their findings are generalizable such that they will be relevant to all women seeking plastic surgery, and (b) that these women will encounter their research, allowing it to influence or inform their decisions. This claim should be tempered or preferably removed from the manuscript.

The authors are very loose with the terminology of non-normative bodies and may wish to reconsider this framing. For example, do people of higher weight have non-normative bodies despite constituting most Americans? Further, the authors’ justification for referring to bodies which have undergone surgery as non-normative is not persuasive; if the body looks similar to others as is suggested, it should not be perceived as non-normative.

Relatedly, “higher weight” may not be best terminology here; for example, men typically have “higher weight” than women but are not innately stigmatized as a result. I suggest the authors look to the fat studies literature and indeed adopt the language of “fatness” rather than a euphemism or medicalized terminology. The same applies to the language of “Bigger-bodied” on p.7; this terminology is unclear.

Please expand on how perceptions of humanness influence interpersonal relationships (p. 9). If this is important as the authors suggest, the mechanism should be further elucidated.

Please specify whether gender or sex was measured. Male is used as a descriptor throughout, however, typically when gender is measured the referent would be men. Were participants all cisgender?

Please clarify why the race category for White includes mixed-race people – is there a disadvantage to having a separate category for mixed-race?

The authors should justify why only White women stimuli were chosen, and situate this in a discussion of representation in research. This is particularly relevant given much cosmetic surgery has historically attempted to produce more stereotypically White features.

Please provide a citation for the Justice Sensitivity inventory, or clarify if this was developed for the present study. If this was developed for the present study, please provide additional information on scale development procedures that were undertaken.

Theorizing the link between perceptions of humanness, empathy, and objectification may help to enrich the theoretical setup of the study, particularly given the focus on appearance in the current paper.

p.19 – “the present study demonstrates that negative attitudes toward plastic surgery extend specifically to the act of undergoing plastic surgery itself…” The present study cannot demonstrate this as only intention to undergo plastic surgery was assessed; the act of undergoing it was not. Please rephrase.

Reviewer #5: As a feminist researcher who has undertaken work on the gendered aspects of cosmetic surgeries, I have been asked by the editor to comment specifically as to whether this paper expresses any misogynistic views. I have read the (revised) paper and the reviewers' comments carefully. This paper absolutely does NOT express any misogynistic views. If anything, it is a woman-focussed paper which demonstrates empathy for the plight of women, who are judged wanting if they do not live up to patriarchal appearance standards, and are also judged harshly if they decide to undergo appearance related surgery. This double standard is the exact point of the paper. As feminist cultural theorist & sociologist Ros Gill points out, “Women are never the right age. We are too young, we’re too old. We are too thin, we’re too fat. We wear too much makeup, we don’t wear enough. We are too flashy in our dress, we don’t take enough care. There isn’t a thing we can do that is right” (2007, p.117) . The authors are attempting to challenge the very misogyny that the reviewer is concerned about. The authors' extremely detailed and thorough reply to the reviewer's concerns on this should allay any concerns on this front, to my mind.

Reviewer #6: I did not review the first version of the manuscript.

The authors seem to have responded appropriately to a series of issues raised by the other reviewers, except maybe one point.

Moreover, I have some points that the authors might want to address.

1) consequences of perception on women's well-being. Although I understand it might be an issue that deserves to be empirically investigated, I do not think that the authors should include a sentence like "we concluded attractive women seeking cosmetic surgery are subject to experience negative psychosocial outcomes" in the abstract. This is too speculative. Same comment for the part at the end of the Low warmth and Competence paragraph. It might be better to have these hypotheses put at the end of the manuscript in the Implications and Future Directions section.

2) attractiveness ratings. Although the photographs were pretested in terms of attractiveness, it could have been better to ask participants to rate the targets’ attractiveness, at least for manipulation check. Moreover, it could have been interesting to see whether planning cosmetic surgery could influence attractiveness ratings and, more importantly, whether it could influence the link between attractiveness and personality inferences.

3) regarding the gender effect point raised by one reviewer, I understand the authors do not want to include this point in the manuscript because it was not part of their analyses. Still, they could have included the variable as a covariate, as it is done in many research in which gender could have potential effects.

4) I understood that there were no effects of Justice Sensitivity. But before that, unless I missed it, the authors did not justify the choice of assessing only the Observer's perspective. One could also imagine that the Victim's perspective, for example, could have been a moderator (women feeling less attractive than the target would not want the woman seeking to be more attractive through cosmetic surgery to be even more favored considering that attractiveness can sometimes be a social advantage).

5) The authors mentioned the what-is-beautiful-is-good stereotype considered as the typical example of a halo effect. Do the authors consider that the results t

7. PLOS authors have the option to publish the peer review history of their article (what does this mean?). If published, this will include your full peer review and any attached files.

Reviewer #1: No

Reviewer #2: No

Reviewer #4: No

Reviewer #5: **Yes: **Dr Paula Singleton

Reviewer #6: No

---

## [Author Response · Author response to Decision Letter 1]

26 Jun 2021

Response to Reviewers

R1.1 Line 258 -259: Please put in () the sentence “ A full list…”.

“Again, each participant would read four of these vignettes in total; three control vignettes (of the 12 in total) and the plastic surgery vignette (vignettes can be accessed on the Open Science Framework; https://tinyurl.com/ska2qv9). Further details are outlined in the procedure.” p11

R1.2 Statistical Analyses: Include in the text, the assumptions criterion used for LMM, according to your reviewer answer.

We appreciate the reviewer’s suggestion to provide assumption checking criteria. Ultimately, we felt it more appropriate to include these criteria on the Open Science Framework (versus in-text). We have noted in-text that assumption checking information can now be found on the Open Science Framework. 

“All data and a step-by-step guide for our statistical analyses (including data cleaning and assumption checking) are available on the Open Science Framework (https://tinyurl.com/ska2qv9).” p15

R1.3 Pg 16, - Pg 16, Line 350 -352: Please put in () the sentence “ Betas…ska2..”.

“However, participant scores for justice sensitivity and disgust sensitivity did not moderate these relationships (betas and confidence intervals for these non-significant interactions are accessible using our step-by-step guide on the Open Science Framework; https://tinyurl.com/ska2qv9).” p15

R1.4 Pg 19, Line 381 -383: Please put in () the sentence “ Betas…ska2..”.

“Participant scores for justice sensitivity and disgust sensitivity did not moderate any of these interactions (betas and confidence intervals for these non-significant interactions are accessible using our step-by-step guide on the Open Science Framework; https://tinyurl.com/ska2qv9).” p18

R1.5 Pg 28, Line 558 -561: Please cut the sentence “ Specifically…things”, as it is sufficient what was already said above.

We have removed this sentence. 

R2.1 Line 13, on p. 2: The authors should provide some literature evidence to support the assertion that [plastic surgery is increasing in popularity]

We appreciate the reviewer’s comment. However, authors understand not providing references in the abstract to be best practice. If the editor disagrees, we will include references pertaining to plastic surgery popularity (i.e. as noted in the first paragraph of our manuscript; p3) in our abstract. 

Line 157, on p.8: “First proposed in 2002, the Stereotype Content Model proposes”. Stereotype Content Model should be referenced.

We have added a reference for this model.

Line 159, on p.8: The phrase “the way we feel about others is said to depend” should be referenced. Who says this?

The reference at the end of the next sentence incorporated this statement, but we have now reiterated this reference on the same line. 

Line 174 on p. 9: The word “therefore” is redundant.

We have removed this word.

Line 315 on p.15: The word “Results” should be preceded by [The]

We appreciate the reviewer’s comment but feel the word ‘the’ is redundant in this sentence. However, we are happy to edit the sentence if the editor feels it is necessary. 

Line 326 on p. 16: “They were also were rated…” one [were] is redundant

We were unable to locate this typo in the text. 

R4.1 My primary concern with the work pertains to the framing of the implications. I think the authors could grapple more with a few specific issues and I believe that doing so would clarify the feminist intentions of the work. As it is presently written, the paper does seem to suggest that women are responsible for the perceptions/stigmatization from others. In particular, that the practical implications of the work suggest intervention for women seeking plastic surgery (e.g., Without this study, uninformed women may choose to seek out plastic surgery and find themselves surprised and disappointed by the negative psychosocial consequences they face postoperatively) rather than intervention for prejudiced audiences. In my reading, this places the burden on the stigmatized group to change, rather than vying for structural change. The authors are not incorrect to suggest that work of the present nature may help to inform women of potential negative outcomes, however, I think the current framing of these implications is slightly inappropriate. To provide a parallel example: it is broadly understood that fat people face significant stigma, however we do not suggest that we should educate people not to get fat so that they will not have to endure that stigma. Broadly, the work would benefit from more in-depth feminist theorization regarding relevant processes of stigma and prejudice.

We appreciate this comment and agree with the reviewer. We have removed the aforementioned sentence from our introduction and implications are now framed solely as pertaining to better understanding plastic surgery stigma. While we appreciate the reviewer’s suggestion to incorporate further feminist theory and agree it is relevant to the manuscript more broadly, we feel the manuscript would more benefit from an additional discussion of the tangible, real-world implications of stigmatization (as opposed to a discussion of theoretical models). Specifically, we have included a section on the psychosocial outcomes associated with stigma (i.e. “The Negative Impact of Stigma”). We hope this better elucidates some of the implications associated with the present study, and satisfies the reviewer’s concerns. 

“Stigmatized groups face considerable challenges. For example, mental health stigma in the workplace can increase employee’s work-related stress and reduce longevity of employment (19). Similarly, addiction stigma can isolate users from both their social networks and support services (20). Finally, stigmatized sexual minorities are subject to intrusive thoughts and physical symptoms (e.g., diarrhea, faintness, cold, or cough) (21). Thus, there is reason to believe that if plastic surgery is indeed stigmatized, this will adversely impact recipients. Therefore, it is important that we understand whether women who undergo plastic surgery are indeed stigmatized.”p5

R4.2 It may be relevant in the introduction to provide plastic surgery statistics specifically for women – men make up an increasing amount of plastic surgery patients, which is important to acknowledge, but also suggests that the statistics presented in the introduction are not particularly relevant to the current study’s focus on women and plastic surgery.

We have incorporated more women-centric statistics in our introduction. 

“Plastic surgery is particularly popular among women, who account for approximately 87% of all plastic surgery recipients (2). Today, nearly 15 million plastic surgeries per year are performed on women in the US alone – a 169% increase over the past 20 years (3,4). Further, plastic surgery rates increased by more than 20% worldwide between the years 2015 and 2019 (2).” p3

R4.3 p.5 “Without this study, uninformed women may choose to seek out plastic surgery and find themselves surprised and disappointed by the negative psychosocial consequences they face postoperatively”. – this is a very strong claim. First, the authors suggest that stigma surrounding plastic surgery is widespread; to make the present claim, the authors should demonstrate that people are not generally aware of this plastic surgery stigma. Second, the authors seem to be suggesting (a) that their findings are generalizable such that they will be relevant to all women seeking plastic surgery, and (b) that these women will encounter their research, allowing it to influence or inform their decisions. This claim should be tempered or preferably removed from the manuscript.

We agree and have removed this claim from the manuscript. 

R4.4 The authors are very loose with the terminology of non-normative bodies and may wish to reconsider this framing. For example, do people of higher weight have non-normative bodies despite constituting most Americans? Further, the authors’ justification for referring to bodies which have undergone surgery as non-normative is not persuasive; if the body looks similar to others as is suggested, it should not be perceived as non-normative. Relatedly, “higher weight” may not be best terminology here; for example, men typically have “higher weight” than women but are not innately stigmatized as a result. I suggest the authors look to the fat studies literature and indeed adopt the language of “fatness” rather than a euphemism or medicalized terminology. The same applies to the language of “Bigger-bodied” on p.7; this terminology is unclear.

We define non-normative bodies in our manuscript as “those that don’t align with dominant societal perceptions of how bodies ought to look or be (p8). Given the rampant prevalence of weight stigma in American society, we do believe that people who are fat have non-normative bodies for the purpose of this study. 

Secondly, we agree that not all plastic surgery bodies can be labelled non-normative. For example, people who do not disclose their plastic surgery to others and otherwise don’t ‘look’ like they’ve had surgery might not be perceived as non-normative. However, we feel as though if either one of these conditions is not met (i.e. an individual is known to have had surgery and/or others suspect them to have had surgery because of the way that they look), these bodies are subject to be considered non-normative (by our definition of the word). We have amended wording in this section to make this distinction clearer. 

“We therefore posit that those more sensitive to disgust might also express greater plastic surgery stigmatization, given that recipients are also planning to acquire a kind of non-normative body (i.e. one that is is stigmatized because surgically enhanced bodies do not align with perceptions of what bodies ought to be).” p8

Finally, we agree with the reviewer’s comment on our terminology and have adapted this throughout the manuscript. 

R4.5 Please expand on how perceptions of humanness influence interpersonal relationships (p. 9). If this is important as the authors suggest, the mechanism should be further elucidated.

“It is important that we examine humanness in conjunction with the Stereotype Content Model because perceptions of humanness directly influence interpersonal relationships (44). For instance, people who are dehumanized are more often victims of objectification (43) and aggression , and receive less empathy from others (45). Hence, it is imperative that we understand whether plastic surgery recipients are dehumanized.” p9

R4.6 Please specify whether gender or sex was measured. Male is used as a descriptor throughout, however, typically when gender is measured the referent would be men. Were participants all cisgender?

This was an oversight and we have changed terminology throughout our manuscript to reflect gender (e.g., men, women). 

R4.7 Please clarify why the race category for White includes mixed-race people – is there a disadvantage to having a separate category for mixed-race?

The authors felt that recognising multi-racial participants as members of all racial groups with which they identify would mean participant demographics would be more thoroughly represented in the present study. Further, and perhaps more importantly, we felt it appropriate that participants be given the option to include as many or as few races in their identification as they felt appropriate (i.e. we didn’t want to ‘box’ participants into being either White or mixed-race, if perhaps they identified predominantly as White, but also felt alternate categories applied to them). Finally, we presented a free-text options to participants; if they wanted to identify as mixed-race (or anything else that was not listed), they were able to write this in the text box.

R4.8 The authors should justify why only White women stimuli were chosen, and situate this in a discussion of representation in research. This is particularly relevant given much cosmetic surgery has historically attempted to produce more stereotypically White features.

We have explained our rationale in the method. Further, we have commented on the limited generalisability of our findings in the discussion. 

“White women stimuli were chosen because we intuited that the majority of our sample would be White.” p11

“Speaking further to the generalizability of the study, we note that only White plastic surgery stimuli were used. These findings therefore cannot be generalized to people of color; specifically, we are unable to establish whether plastic surgery recipients who are people of color are subject to the negative plastic surgery effect. Given that plastic surgery has historically attempted to produce more stereotypically White features (e.g., surgeries for the ‘Jewish nose’ or ‘Black nose’), it is especially important that we acknowledge the limited applicability of our findings (28,63).” p24

R4.9 Please provide a citation for the Justice Sensitivity inventory, or clarify if this was developed for the present study. If this was developed for the present study, please provide additional information on scale development procedures that were undertaken.

This was an oversight and we have now included this citation. 

R4.10 Theorizing the link between perceptions of humanness, empathy, and objectification may help to enrich the theoretical setup of the study, particularly given the focus on appearance in the current paper.

See R4.5 

R.411 p.19 – “the present study demonstrates that negative attitudes toward plastic surgery extend specifically to the act of undergoing plastic surgery itself…” The present study cannot demonstrate this as only intention to undergo plastic surgery was assessed; the act of undergoing it was not. Please rephrase.

We have rephrased this section and added examples. 

“We posit that exploring plastic surgery as an intended action (versus a completed action) enables researchers to better assess negative perceptions that pertain exclusively to plastic surgery itself, rather than its associated outcomes (i.e. how recipients look after surgery). In other words, studying perceptions of women planning to have plastic surgery allows us to assess shifts in perception regarding recipient character (e.g., “I don’t condone plastic surgery because it is immoral”) as opposed to regarding recipient appearance (e.g., “I don’t condone plastic surgery because I think it makes women look unappealing”). As such, our study aims to establish exactly how a woman’s decision to undergo plastic surgery shapes others’ perceptions of her, irrespective of her surgical outcomes.”p5

R6.1 Consequences of perception on women's well-being. Although I understand it might be an issue that deserves to be empirically investigated, I do not think that the authors should include a sentence like "we concluded attractive women seeking cosmetic surgery are subject to experience negative psychosocial outcomes" in the abstract. This is too speculative. Same comment for the part at the end of the Low warmth and Competence paragraph. It might be better to have these hypotheses put at the end of the manuscript in the Implications and Future Directions section.

We agree with the reviewer. We have reworded the aforementioned sentence in the abstract to better reflect that the authors can only speculate as to how cosmetic surgery influences psychosocial outcomes. We have also reframed the end of our warmth and competence paragraph. Finally, and most significantly, we have noted in our future directions that these outcomes are speculative only, and that future research is needed to confirm whether the negative plastic surgery effect induces contempt and worsened wellbeing. 

“As such, we contend that attractive women seeking plastic surgery may find themselves experiencing negative psychosocial outcomes; they may be interpersonally and professionally affected by others’ feelings of contempt toward them and may also be less likely to fulfil their basic psychological needs. However, we note that at present these outcomes are purely speculative, and that future research is needed to test these associations.” p4

R6.2 Attractiveness ratings. Although the photographs were pretested in terms of attractiveness, it could have been better to ask participants to rate the targets’ attractiveness, at least for manipulation check. Moreover, it could have been interesting to see whether planning cosmetic surgery could influence attractiveness ratings and, more importantly, whether it could influence the link between attractiveness and personality inferences.

We appreciate the reviewer’s feedback. We agree that including a manipulation check would’ve been beneficial and will apply this suggestion to any future research we conduct using these stimuli. However, given that the Chicago Faces Database has previously undergone rigorous and cross-cultural validity assessments, we remain confident that the manipulation was successful (especially considering that we selected the eight most and eight least attractive faces for our stimuli). We also agree that it would be interesting to assess whether cosmetic surgery influences stimuli attractiveness ratings and the relationship between attractiveness ratings and personality inferences. While we are presently unable to conduct this analysis (given that we did not collect attractiveness ratings), we will include this analysis if we conduct similar research in the future. 

6.3 Regarding the gender effect point raised by one reviewer, I understand the authors do not want to include this point in the manuscript because it was not part of their analyses. Still, they could have included the variable as a covariate, as it is done in many research in which gender could have potential effects.

As previously reported in our first revision letter, with the exception of the single morality outcome, participant gender did not moderate the effect of plastic surgery or participant attractiveness on judgements of the stimuli. Additionally, gender did not have a significant main effect on ratings on our outcome measures. We did not have any theoretical reason to expect gender to play an important role in our study, and given the non-significant impact of gender in our analyses, including it as a covariate would not change any reported results and would, in our opinion, add little to the reported story. We acknowledge that readers may raise a similar question as to the potential impact of gender, and so to address this we have added a sentence to the body of our results summarising the (lack of) gender effects in our study.

“At the request of a reviewer, we also examined whether any of the aforementioned results were moderated by participant gender. We found that only the relationship between plastic surgery condition and morality but not warmth was moderated by gender, such that the relationship was only significant when participants were women. All other analyses were unaffected by gender and thus it was not included in reported models.” p19

6.4 I understood that there were no effects of Justice Sensitivity. But before that, unless I missed it, the authors did not justify the choice of assessing only the Observer's perspective. One could also imagine that the Victim's perspective, for example, could have been a moderator (women feeling less attractive than the target would not want the woman seeking to be more attractive through cosmetic surgery to be even more favored considering that attractiveness can sometimes be a social advantage).

We agree with the reviewer that other subscales of the Justice Sensitivity Inventory (JSI) could’ve acted as a moderator for the negative plastic surgery effect. However, when conceptualising the study, we ultimately felt that the observer subscale of the JSI subsumed both self-oriented and other-oriented feelings of injustice; for example, items such as “I am upset when someone does not get a reward he/she has earned” can reflect either oneself as ‘someone’ or an external individual as ‘someone’. Conversely, items on the victim subscale (i.e. the subscale noted by the reviewer) solely apply to wrongdoing towards oneself. As such, we chose to use the observer subscale because it incorporated several different possible experiences of perceived injustice and we therefore felt it more comprehensive. 

6.5 The authors mentioned the what-is-beautiful-is-good stereotype considered as the typical example of a halo effect. Do the authors consider that the results t …

Unfortunately, the second half of this reviewer comment was cut from the email. We therefore weren’t sure how to address this comment.

---

## [Editor Report · Decision Letter 2]

5 Jul 2021

PONE-D-21-05038R2

Under the Knife: Unfavorable Perceptions of Women Who Seek Plastic Surgery

PLOS ONE

Dear Dr. Bonell,

Thank you for submitting your manuscript to PLOS ONE. Parts of comment 5 by Reviewer 6 were missing. So, I contacted the reviewer, who cordially has just reverted with the full comment (see below).

Comment 5: *"5) The authors mentioned the what-is-beautiful-is-good stereotype considered as the typical example of a halo effect. Do the authors consider that the results they found could be considered as an example of a horn effect?"*

Therefore, I invite you to submit a revised version of the manuscript that addresses the points raised during the review process in full, including a response to my comments on how the peer-review was managed to address any concerns about misogynistic sentiments in the text.

We look forward to receiving your revised manuscript.

Kind regards,

Ali B. Mahmoud, Ph.D.

Academic Editor

PLOS ONE
---

## [Author Response · Author response to Decision Letter 2]

5 Jul 2021

Response to Reviewers – Review Round 1

Reviewer 1

R1.1 Miscellaneous

Although it was highlight, that perceptions of attractive women are worsened when these women decide to seek cosmetic surgery, authors only refer that psychoeducation content can be developed for women considering plastic surgery, without specifying which ones.

We have made clearer throughout our manuscript the practical implications that may arise from our findings. Namely, we have shifted focus away from the generation of psychoeducation content. Instead, we focus on how the present findings might shape our understanding of plastic surgery attitudes, and how women who might be considering plastic surgery can benefit from our findings. We feel these implications are more realistic and more in line with the aims of the present study. See changes throughout our manuscript and see below for an example of these changes.

e.g., “This study is one of few that elucidates the potential psychosocial challenges women seeking plastic surgery might face. As such, findings from this study might prove informative over and above any information women might receive from surgeons or surgical clinics (i.e. where the focus in on physical outcomes only) and/or might work to educate surgeons themselves so that they are better able to inform their clientele. Specifically, we feel that from an ethical standpoint it is important for women seeking cosmetic surgery to fully understand the implications of their decisions (e.g., in undergoing cosmetic surgery, they may be perceived as less human, among other things).” p25

R1.2 Title

Should be written in sentence case (only the first word of the text, proper nouns, and genus names are capitalized).

We applied this revision. 

“Under the knife: Unfavorable perceptions of women who seek plastic surgery” p1

R1.3 Abstract

R1.3.1 Line 15: “….985 adults…” – what kind of sample (random, convenience)? Please explain.

We have specified that our sample was random. 

“We presented a random sample of 985 adults (male = 54%, Mage = 35.84 years) recruited via Amazon’s Mechanical Turk with a series of experimental stimuli” p2

R1.3.2 Line 16: “…M age…” Please add the standard deviation.

We have added standard deviation 

“(male = 54%, Mage = 35.84 years, SDage = 10.59)” p2

R1.3.3 Line 19: “…humanness”. In the end of the sentence, please refer the statistical analyses performed.

We have included our analysis. 

“Participants rated stimuli on perceived warmth, competence, morality, and humanness. We ran linear mixed-effect models to assess all study hypotheses.”p2

R1.3.4 Line 21: “… control activities …” Please specify/give examples.

We feel that adding more information pertaining to the control activities in the abstract would confuse readers more so than aid clarity. However, we have added an additional example of for control activities in the Materials and Measures section of our Method. An exhaustive list of our control vignettes has now also been uploaded to the OSF, and this has been indicated in our manuscript. 

“Of these, 12 were control vignettes that depicted a woman planning to undergo a neutral – that is, common or everyday – activity (e.g., “this woman is planning to eat a meal”, “this woman is planning to buy a pet”). The remaining vignette was our plastic surgery target vignette – “this woman is planning to have plastic surgery”. Again, each participant would read four of these vignettes in total; three control vignettes (of the 12 in total) and the plastic surgery vignette. Further details are outlined in the procedure. A full list of control vignettes can be found on the Open Science Framework (https://tinyurl.com/ska2qv9).” p11

R1.4 Introduction

R.1.4.1 Despite being presented, in theoretical terms, the contextualization of the problem, it is not elaborated, in practical terms, what this investigation can contribute to the resolution of the problem. Please explain to what extent the perception of others about plastic surgery is important and how it may have practical implications for those who intend to perform plastic surgery.

See our response to R1.1 for an overview of how we included more practical implications in our discussion. For our introduction, we also included this line:

“This knowledge will help contribute to our growing understanding of how society perceives plastic surgery, thereby elucidating the challenges women who choose to undergo it are likely to face. Without this study, uninformed women may choose to seek out plastic surgery and find themselves surprised and disappointed by the negative psychosocial consequences they face postoperatively.” p5

R1.4.2 Line 67: “…Brazilian women”. Please cite reference.

We have added three references.

“…power for Brazilian women; 13–15).” p4

R1.4.3 Line 84 – “…psychoeducation content…”: Please elaborate / specify.

This line has been removed from the manuscript, rendering this suggestion obsolete. See R1.1 for removal justification. 

R.1.3.4 Lines 173-175; 177-178: Very confusing phrases: after all you intend to evaluate the perception of women who intend to have a plastic surgery or the perception of others about women who intend to have plastic surgery? Please clarify since it is incongruous with the presented hypotheses.

We have reworded these lines. 

“In an attempt to examine the social consequences of undergoing plastic surgery, the present paper will therefore examine how women who seek plastic surgery are perceived across four domains: warmth, competence, morality, and humanness … We aimed to examine whether perceptions of women who seek plastic surgery systematically differ from perceptions of women who do not.”p9

R1.3.5 Since the participants were from the two sex, it would be interesting to explore if women seeking plastic surgery are perceived by males and females in the same way.

We examined the effect of participant gender on the negative plastic surgery effect. While effects were non-significant for most outcome variables (warmth, competence, and humanness), we did find that plastic surgery status only predicted morality scores in cases where participants were female. In other words, women seeking plastic surgery were only considered less moral than those not seeking surgery by female participants.

While ultimately we did not decide to include this exploratory analysis in our revised manuscript (given that it largely produced non-significant results and was not part of our initial analysis plan), we do value improving our understanding of these variables and their interactions and thank the reviewer for their suggestion. 

R1.5 Material and Methods

R1.5.1 Line 196: Explain how the sample size was calculated. If a sample size calculation was performed, specify the inputs for power, effect size and alpha.

No sample size calculation was performed for this study. In hindsight, we understand that ideally a sample size calculation would have been performed. However, in the interest of transparency, we note that sample size was informed exclusively by funding available to our lab at the time of data collection. We have indicated this in the manuscript. 

“Sample size was determined prior to data analysis based on funding available to our research team at the time of data collection.”p10

R1.5.2 Lines 202-203: ”… see …data quality). This sentence should be cut and only the reference should appear. In limitation you should refer this.

We have amended this sentence. We do not believe our compensation to be a limitation; rather, we were attempting to refer readers to this article in case they were wondering why we compensated participants more generously than most other studies on MTurk. 

“Compensation for each participant completing our 15-minute survey was US$2.33 (42).”p10

R1.5.3 Lines 206-207: ”… see …checks). This sentence should be cut and only the reference should appear. Justify in the text, why you excluded 16 participants and cite the reference.

We have amended this. 

“We excluded 16 participants from the present study for failing to consistently report their race. To elaborate, we included two items in our survey asking participants to indicate their race – one toward the commencement, and one toward the conclusion, of the survey. In cases where participant responses did not align between these two items, ‘participants’ were assumed to be bots (or to simply not be paying attention) and were subsequently removed from the study (43).” p10

R1.5.4 Line 227: How many participants for the pilot study?

208. We have included this in the manuscript. This information is also available in the data file on the Open Science Framework. 

“All vignettes were assessed for their ability to induce participant affect and arousal in an MTurk pilot study (N = 208)” p11

R1.5.5 What is the assumptions criterion used for LMM?

Before interpreting the output of any multilevel models, we checked the key assumptions of linearity of relationships (where relevant, as many of our analyses had binary predictors only, thus linearity was inherent), normality of residuals, and homoscedasticity. For these checks, we took a graphical approach as recommended by Fife (2020). We used the R-packagesjPlot (Lüdecke,2020) to draw diagnostic plots. To check for linearity, we plotted model residuals against predictors. To check for normality in fixed effects, we plotted a Q-Q plot of residuals and the distribution of residuals. To check for normality in random effects, we plotted random effect quantiles against standard normal quantiles. To check for homoscedasticity, we plotted fitted values against residuals. All assumptions fell within reasonable bounds upon visual inspection.

R1.5.6 Describe any analysis carried out to confirm the data meets the assumptions of the analysis performed (e.g.: linearity, co-linearity, normality of the distribution).

See R1.5.5

R1.5.7 Please explain in detail how was made the data generation (Describe the technical details or procedures required to reproduce the analysis)

This is available on the Open Science Framework. We had noted this in our manuscript, but have also amended the wording to make it clear that a full guide is provided on the Open Science Framework. 

“All data and a step-by-step guide for our statistical analyses (including data cleaning) are available on the Open Science Framework (https://tinyurl.com/ska2qv9).”p15

R1.5.8 Please indicate what was the statistical package software used to carry out LMM analyses (List the name and version of software package used, alongside any relevant references).

RStudio Version 1.2.1335 was used.

“All analyses were conducted in RStudio Version 1.2.1335 (48).”p15

R1.6 Results

R1.6.1 Lines 313-314: “...these interactions). Please indicate the ß and CI 95%.

ß and CI 95% can be calculated using the step-by-step guide we provide on the Open Science Framework. We feel as though providing all ßs and CIs in our manuscript would add unnecessary confusion and bulk to our text. However, we have added a note in our manuscript that these statistics are accessible using the Open Science Framework. 

“Participant scores for justice sensitivity and disgust sensitivity did not moderate any of these interactions. Betas and confidence intervals for these non-significant interactions are accessible using our step-by-step guide on the Open Science Framework (https://tinyurl.com/ska2qv9).” p15

R1.6.2 Line 326: “…were also were…” cut the second “were”.

We’ve cut out this typo. 

R1.6.3 Lines 340-341: “...these relationships). Please indicate the ß and CI 95%.

See R1.6.1 

“However, participant scores for justice sensitivity and disgust sensitivity did not moderate these relationships. Betas and confidence intervals for these non-significant interactions are accessible using our step-by-step guide on the Open Science Framework (https://tinyurl.com/ska2qv9).” p18

R1.7 Discussion

R1.7.1 Lines 349-350: Again, very confusing phrase: you intend to evaluate the perception of women who intend to have a plastic surgery or the perception of others about women who intend to have plastic surgery? Please clarify

Again, we have amended our wording.

“The present study built on existing literature by examining whether women seeking plastic surgery are systematically perceived differently to other women.” p19

R1.7.2 Line 353: “…control activities”: Please give examples.

Given that we have reiterated that control activities can be found on the Open Science Framework, as well as included more examples in our method section (see R1.3.4), we don’t feel it necessary to here again include examples of control activities. However, if the reviewer ultimately disagrees, we will happily amend to include the discussion to include examples. 

R1.7.3 Line 369: “…is unique” Please cite references.

We have included a reference for this sentence.

“…has demonstrated that the way in which people relate to members of each quadrant is unique (37).” p19

R1.7.4 Lines 376-377: “…seeking plastic surgery is likely to foster implication for women …”. I wonder, although this is a possibility in the face of the choice of a woman to decide to have plastic surgery, something that in principle should only concern her, the fact that she does not do it for the sake of others, will have milder or more serious psychosocial implications/impact for the own? Please elaborate about it.

We agree that it would be interesting to investigate whether women’s motivations for undergoing surgery (i.e. for themselves versus for others) would affect how the negative plastic surgery effect might impact their psychosocial wellbeing. However, because this area of research is so underexamined, we feel there are an abundance of potential, unexplored variables that might moderate this relationship (e.g., surgical outcome, recipient gender, recipient age etc). As such, we feel it is outside the scope of our paper to address all of these moderators, and would prefer not to incorporate one in lieu of others. 

R1.7.5 Line 385: “…inform our own”. Please cite reference.

This line is already cited – perhaps this was an accidental oversight by the reviewer.

“In line with socialization theory (whereby the beliefs of those around us heavily inform our own), we induce that women considering plastic surgery may struggle to ‘feel moral’ if others perceive them to be morally questionable (56).” p20

R1.7.6 Line 387: “…wellbeing relative to …”. Relative to what?? Please explain.

Again, we believe this might be a reviewer oversight, as we explain this is relative to if they were not seeking plastic surgery. 

“In turn, they may experience worsened wellbeing relative to if they were not seeking plastic surgery.” p20

R1.7.7 Line 394: “…and that this in turn …”. Cut “that this”.

We believe the original wording of the sentence to be grammatically correct, but added a comma for clarity.

“Overall, we conclude that the ability to fulfil psychological needs may be affected for women seeking plastic surgery, and that this in turn may detriment their wellbeing.” p20

R1.7.8 Lines 400-401: “…was not …by results”. Why? What is the possible explanation? Please elaborate.

We have expanded on this section. See R2.5.

R1.7.9 Lines 417-419: “As such, …surgery effect”. Why? What is the possible explanation? Please elaborate.

Ultimately, given that this was a purely exploratory hypothesis grounded in little existing literature or theory, we have decided not to expand on our interpretation of this result. However, we are happy to make further changes and expand on this section if the reviewer is not happy with our decision. 

R1.7.10 Lines 426, 435, 449: “To elaborate”. Please avoid the systematic repetition of the expression.

We removed one of these sentences from our manuscript, and reworded another to avoid repetition.

R1.7.11 Lines 460-463: The author refers to women perception who intend to have a plastic surgery or the perception of others about women who intend to have plastic surgery? Please clarify!

We have reworded this sentence to aid clarity. 

“In other words, while there were statistically significant differences between perceptions of women who seek surgery and women who do not across all outcome measures, the absolute difference in perceptions of these women were minimal.” p24

R1.7.12 Limitation: As you refer on the participants section, it was made a compensation for each participant completing the survey. Taking this in account, how do you guarantee honest responses?

Providing compensation to participants is generally considered best practice, so long as payment amount is appropriate (i.e. not so large that it is deemed coercive or undue; e.g., see Ripley [2006] for a review). While we cannot guarantee honest responses, we did include an attention check measure in our study to ensure ‘bots’ and distracted workers’ responses were not included in our analyses (see R1.5.3). 

R1.7.13 Line 471: “…psychoeducation content…”. Please elaborate / specify.

We have removed this line. See R1.1.

R1.7.14 Lines 473-475: “… we suspect…planning to do so”. I am confused… taking in account that this study “… posits that exploring plastic surgery as an intended action would enable researchers to better assess negative perceptions …rather than its associated outcomes”, how do you explain your suspicious??

We agree with the reviewer’s concerns regarding this section, and have subsequently removed it from our discussion.

R1.7.15 Lines 478-479: “Overall…experimentally”. By whom?? Please refer.

We suspect the reviewer has misread this sentence in our manuscript, as we are referring to our own study at this point (see below). However, we are happy to revisit the comment if we have misunderstood. 

“Overall, our study was the first to examine the negative plastic surgery effect experimentally.” p25

R1.7.16 Lines 479-480: “we…attitudes”. Please highlight the importance of the thematic, concerning future studies.

We weren’t too sure what the reviewer meant by ‘thematic’, but we have attempted to address this feedback by including more information about future implications.

“We provide a fundamental starting point from which future literature can further investigate negative plastic surgery attitudes in order to inform both women seeking cosmetic surgery and cosmetic surgeons themselves.” p25

R1.8 References/Citations

If the intention is to use Vancouver style consistently, please revise its correct use and format throughout the text. Special attention should be given to citations of references (e.g., Lines: 56, 74, 100, 124, 135, 151, 163, 203 (this one is in APA style), 207, 215…).

Please revise “DOI” number and adhere to the format give in “PLOS ONE Submission Guidelines”.

We have removed page numbers (e.g., line 56). For lines 74, 100, 124, 135, 151, 163, and 215, bracketing has been corrected. We have amended lines 203 and 207. Further, as per PLoS ONE guidelines, DOI numbers can be used in place of page numbers but are not necessary for inclusion if page numbers are already included. As such, we have not added DOIs to our references. 

R1.9 Tables

The titles should not be in italic and should be align with table identification.

We have amended this formatting. 

Reviewer 2

R2.1 Abstract and Introduction: Here I suggest that the authors provide some definition of the terms ‘attractiveness’ and ‘unattractiveness’. If these are major construct unpinning the perceptions that they are seeking to explain, then they should describe what they mean by these terms. It may be that they see attractiveness and unattractiveness from the participants’ subjective appreciation/judgement, but then the authors have to explain that.

We have added a brief note in our introduction to make clear what we mean by attractiveness. Also, we have reiterated how attractiveness was conceptualised in the present study in the materials and measures section of our method. 

“An individual’s physical attractiveness – by which we mean their average attractiveness score as rated by others – affects how others perceive them.” p5

“We compiled a smaller database containing the eight most and eight least attractive (as rated by the 1087 CFD participants) White women contained in the CFD for use in the present study (i.e. 16 photographs in total).” p11

R2.2 Introduction (p.3, line 36). The authors claim that plastic surgery demand and operations have increased significantly. It will be helpful if they provide some data from the literature or health to substantiate this. Also, they could define a broad time period over which such increase can be observed.

We agree, and have made the necessary revisions. 

“Today, approximately 18 million plastic surgeries are performed in the US each year – a 169% increase since the year 2000 (2,3). Further, rates increased by more than 20% worldwide between the years 2015 and 2019 (4).” p3

R2.3 Hypotheses (p.9). Overall, it would be helpful if the authors number their hypotheses for ease of reference from a reader perspective but also for themselves to recapture them in the discussion or conclusion. The hypotheses on the same p.9 (line…) is not clearly related to the research question and the aim of the study which to examine the perception of women seeking plastic surgery or having had plastic surgery. I do see the connection. Perhaps the authors could explain and establish the link clearly.

We have numbered our hypotheses (1, 2, 3, 4a, 4b) for ease of reference. We were not entirely sure whether the editor’s comments pertaining to ‘the hypotheses on the same p.9 (line…)’ referred to a specific hypothesis on p9, or rather our hypotheses in general. However, we posit that all our hypotheses bar hypothesis 3 are directly related to examining the perception of women seeking plastic surgery. Therefore, we assume the reviewer is referring to hypothesis 3. In light of this, we contend that hypothesis 3, while not directly related to our manuscript’s research question, works toward validating existing literature (as stated in the manuscript). Therefore, we feel its inclusion is justified. We hope this response has satisfied the reviewer’s concerns and are happy to make future changes to the manuscript if necessary. 

“Our primary hypothesis was that (1) women seeking plastic surgery would be considered less warm, competent, moral, and human than those who are not (i.e. there would be a negative plastic surgery effect). We also hypothesized that (2) this relationship… (3)… (4a)… (4b)”p9

R2.4 Methods (p.10). The word [in] might be missing before [total] in the following phrase (i.e. 16 photographs total). Same for line 224 on p.11 in the phrase (of the 16 total) and line 270 on p.13 in the phrase (16 items total). Please check these parentheses.

We have applied these revisions to all noted parentheses. 

R2.5 Results. On p.20, the coverage of Justice sensitivity needs elaboration drawing of the data and contrasting with the literature. 

We have elaborated in this section.

“Firstly, contrary to existing literature, these findings do not support the theory that the negative plastic surgery effect is driven by concerns pertaining to recipients reaping ‘unearned’ rewards from plastic surgery (i.e. the concern that plastic surgery is 'cheating'; 11,25,55). Further, also in contrast with existing literature, these findings also do not support the theory that the negative plastic surgery effect is driven by stigmatization toward individuals with non-normative bodies (22,26,27). We note, however, that because of our study design, stimuli described as having had plastic surgery in the present study did not necessarily look non-normative (i.e. they did not look different to control stimuli). As such, we cannot definitively conclude that perceptions of disgust do not drive the negative plastic surgery effect in cases where women have visibly undergone surgery (e.g., they look ‘artificial’); rather, only in cases where plastic surgery status alternatively becomes known.” p21

R2.6 Implications. The authors could elaborate on the practical implications of the study under the heading on p.20. 

See R1.1. We have incorporated our implications into a section titled “Implications, Conclusions, and Future Directions” on page 24. 

R2.7 Future research. I think it would interesting for future research to also look at the consequences of such negative perception of ‘attractive’ women undergoing plastic surgery. Does it lead to physical aggression? Stereotyping? Some forms of exclusion? And what those women affect could do?

We have incorporated this suggestion in the final section of our manuscript. 

“Future research might also explore additional consequences that women subject to the negative plastic surgery effect are likely to face. For example, might this worsened person perception result in the social exclusion or prejudicial treatment?” p24

Reviewer 3

The premise of this research and the methods used (i.e. rating women on attractiveness) is steeped in misogyny and has no place in contemporary scholarship. We must do better. See this media piece for a summary of the types of issues that are promoted by your work: https://www.theguardian.com/society/2020/aug/05/disgusting-study-rating-attractiveness-of-women-with-endometriosis-retracted-by-medical-journal

We appreciate and share reviewer 3’s disdain for the ongoing misogyny rife in contemporary scholarship. Further, we are familiar with the endometriosis study provided as an example by reviewer 3 and also believe this work to be fundamentally problematic. However, we disagree with reviewer 3 with regards to the role our own manuscript plays in promoting misogyny in contemporary research. Notably, while we (assumedly) agree with reviewer 3 in that women should not be judged based on their attractiveness, we feel it is important to acknowledge that they nevertheless are (see Dion’s 1972 seminal “What is beautiful is good”; see commentary on ‘lookism’ by Australian scholar Beatrice Alba – https://tinyurl.com/4fd4azbp). The intent of our manuscript is not to encourage the objectification of women, but rather (in part) to investigate the very real and important role attractiveness plays in person perception. 

Dissimilarly to the endometriosis study referenced by reviewer 3, we feel our manuscript has clear scientific merit. For example, our study works to validate existing literature as well as anecdotal accounts of women’s lived experiences of discrimination based on ‘lookism’ (see commentary above). Further, our study importantly examines whether women seeking cosmetic surgery – potentially to avoid facing discrimination based on their attractiveness – might actually be subject to further discrimination for choosing to undergo surgery. Finally, we note that at no stage during our study were participants actually asked to rate each other (or even photographed stimuli) on attractiveness. Rather, our methodology relied on a set of pre-existing, validated faces already rated on attractiveness (the Chicago Faces Database; CFD). Importantly, CFD participants provided full informed consent to be rated on attractiveness. 

Lastly, we would like to make known to reviewer 3 that all collaborators on this paper are members of what we consider to be a feminist research team. While we may not agree with reviewer 3’s concerns regarding our work, we very much do appreciate the opportunity to engage in feminist dialogue and recognise that differences in opinion will exist even within the feminist community. In an attempt to placate some of reviewer 3’s concerns, we have added a brief note in our introduction to make clear what we mean by ‘attractiveness’ in this context (thereby eliminating any misconceptions that our research team believes beauty to be an entirely objective or dichotomous construct). Further, we have reiterated how attractiveness was conceptualised in the present study in the materials and measures section of our method. We hope that this response has addressed reviewer 3’s concerns; however, we are of course happy to take on board any further suggestions for improving our manuscript. 

“An individual’s physical attractiveness – by which we mean their average attractiveness score as rated by others – affects how others perceive them.” p5

“We compiled a smaller database containing the eight most and eight least attractive (as rated by the 1087 CFD participants) White women contained in the CFD for use in the present study (i.e. 16 photographs in total).” p11

Response to Reviewers – Review Round 2

R1.1 Line 258 -259: Please put in () the sentence “ A full list…”.

“Again, each participant would read four of these vignettes in total; three control vignettes (of the 12 in total) and the plastic surgery vignette (vignettes can be accessed on the Open Science Framework; https://tinyurl.com/ska2qv9). Further details are outlined in the procedure.” p11

R1.2 Statistical Analyses: Include in the text, the assumptions criterion used for LMM, according to your reviewer answer.

We appreciate the reviewer’s suggestion to provide assumption checking criteria. Ultimately, we felt it more appropriate to include these criteria on the Open Science Framework (versus in-text). We have noted in-text that assumption checking information can now be found on the Open Science Framework. 

“All data and a step-by-step guide for our statistical analyses (including data cleaning and assumption checking) are available on the Open Science Framework (https://tinyurl.com/ska2qv9).” p15

R1.3 Pg 16, - Pg 16, Line 350 -352: Please put in () the sentence “ Betas…ska2..”.

“However, participant scores for justice sensitivity and disgust sensitivity did not moderate these relationships (betas and confidence intervals for these non-significant interactions are accessible using our step-by-step guide on the Open Science Framework; https://tinyurl.com/ska2qv9).” p15

R1.4 Pg 19, Line 381 -383: Please put in () the sentence “ Betas…ska2..”.

“Participant scores for justice sensitivity and disgust sensitivity did not moderate any of these interactions (betas and confidence intervals for these non-significant interactions are accessible using our step-by-step guide on the Open Science Framework; https://tinyurl.com/ska2qv9).” p18

R1.5 Pg 28, Line 558 -561: Please cut the sentence “ Specifically…things”, as it is sufficient what was already said above.

We have removed this sentence. 

R2.1 Line 13, on p. 2: The authors should provide some literature evidence to support the assertion that [plastic surgery is increasing in popularity]

We appreciate the reviewer’s comment. However, authors understand not providing references in the abstract to be best practice. If the editor disagrees, we will include references pertaining to plastic surgery popularity (i.e. as noted in the first paragraph of our manuscript; p3) in our abstract. 

Line 157, on p.8: “First proposed in 2002, the Stereotype Content Model proposes”. Stereotype Content Model should be referenced.

We have added a reference for this model.

Line 159, on p.8: The phrase “the way we feel about others is said to depend” should be referenced. Who says this?

The reference at the end of the next sentence incorporated this statement, but we have now reiterated this reference on the same line. 

Line 174 on p. 9: The word “therefore” is redundant.

We have removed this word.

Line 315 on p.15: The word “Results” should be preceded by [The]

We appreciate the reviewer’s comment but feel the word ‘the’ is redundant in this sentence. However, we are happy to edit the sentence if the editor feels it is necessary. 

Line 326 on p. 16: “They were also were rated…” one [were] is redundant

We were unable to locate this typo in the text. 

R4.1 My primary concern with the work pertains to the framing of the implications. I think the authors could grapple more with a few specific issues and I believe that doing so would clarify the feminist intentions of the work. As it is presently written, the paper does seem to suggest that women are responsible for the perceptions/stigmatization from others. In particular, that the practical implications of the work suggest intervention for women seeking plastic surgery (e.g., Without this study, uninformed women may choose to seek out plastic surgery and find themselves surprised and disappointed by the negative psychosocial consequences they face postoperatively) rather than intervention for prejudiced audiences. In my reading, this places the burden on the stigmatized group to change, rather than vying for structural change. The authors are not incorrect to suggest that work of the present nature may help to inform women of potential negative outcomes, however, I think the current framing of these implications is slightly inappropriate. To provide a parallel example: it is broadly understood that fat people face significant stigma, however we do not suggest that we should educate people not to get fat so that they will not have to endure that stigma. Broadly, the work would benefit from more in-depth feminist theorization regarding relevant processes of stigma and prejudice.

We appreciate this comment and agree with the reviewer. We have removed the aforementioned sentence from our introduction and implications are now framed solely as pertaining to better understanding plastic surgery stigma. While we appreciate the reviewer’s suggestion to incorporate further feminist theory and agree it is relevant to the manuscript more broadly, we feel the manuscript would more benefit from an additional discussion of the tangible, real-world implications of stigmatization (as opposed to a discussion of theoretical models). Specifically, we have included a section on the psychosocial outcomes associated with stigma (i.e. “The Negative Impact of Stigma”). We hope this better elucidates some of the implications associated with the present study, and satisfies the reviewer’s concerns. 

“Stigmatized groups face considerable challenges. For example, mental health stigma in the workplace can increase employee’s work-related stress and reduce longevity of employment (19). Similarly, addiction stigma can isolate users from both their social networks and support services (20). Finally, stigmatized sexual minorities are subject to intrusive thoughts and physical symptoms (e.g., diarrhea, faintness, cold, or cough) (21). Thus, there is reason to believe that if plastic surgery is indeed stigmatized, this will adversely impact recipients. Therefore, it is important that we understand whether women who undergo plastic surgery are indeed stigmatized.”p5

R4.2 It may be relevant in the introduction to provide plastic surgery statistics specifically for women – men make up an increasing amount of plastic surgery patients, which is important to acknowledge, but also suggests that the statistics presented in the introduction are not particularly relevant to the current study’s focus on women and plastic surgery.

We have incorporated more women-centric statistics in our introduction. 

“Plastic surgery is particularly popular among women, who account for approximately 87% of all plastic surgery recipients (2). Today, nearly 15 million plastic surgeries per year are performed on women in the US alone – a 169% increase over the past 20 years (3,4). Further, plastic surgery rates increased by more than 20% worldwide between the years 2015 and 2019 (2).” p3

R4.3 p.5 “Without this study, uninformed women may choose to seek out plastic surgery and find themselves surprised and disappointed by the negative psychosocial consequences they face postoperatively”. – this is a very strong claim. First, the authors suggest that stigma surrounding plastic surgery is widespread; to make the present claim, the authors should demonstrate that people are not generally aware of this plastic surgery stigma. Second, the authors seem to be suggesting (a) that their findings are generalizable such that they will be relevant to all women seeking plastic surgery, and (b) that these women will encounter their research, allowing it to influence or inform their decisions. This claim should be tempered or preferably removed from the manuscript.

We agree and have removed this claim from the manuscript. 

R4.4 The authors are very loose with the terminology of non-normative bodies and may wish to reconsider this framing. For example, do people of higher weight have non-normative bodies despite constituting most Americans? Further, the authors’ justification for referring to bodies which have undergone surgery as non-normative is not persuasive; if the body looks similar to others as is suggested, it should not be perceived as non-normative. Relatedly, “higher weight” may not be best terminology here; for example, men typically have “higher weight” than women but are not innately stigmatized as a result. I suggest the authors look to the fat studies literature and indeed adopt the language of “fatness” rather than a euphemism or medicalized terminology. The same applies to the language of “Bigger-bodied” on p.7; this terminology is unclear.

We define non-normative bodies in our manuscript as “those that don’t align with dominant societal perceptions of how bodies ought to look or be (p8). Given the rampant prevalence of weight stigma in American society, we do believe that people who are fat have non-normative bodies for the purpose of this study. 

Secondly, we agree that not all plastic surgery bodies can be labelled non-normative. For example, people who do not disclose their plastic surgery to others and otherwise don’t ‘look’ like they’ve had surgery might not be perceived as non-normative. However, we feel as though if either one of these conditions is not met (i.e. an individual is known to have had surgery and/or others suspect them to have had surgery because of the way that they look), these bodies are subject to be considered non-normative (by our definition of the word). We have amended wording in this section to make this distinction clearer. 

“We therefore posit that those more sensitive to disgust might also express greater plastic surgery stigmatization, given that recipients are also planning to acquire a kind of non-normative body (i.e. one that is is stigmatized because surgically enhanced bodies do not align with perceptions of what bodies ought to be).” p8

Finally, we agree with the reviewer’s comment on our terminology and have adapted this throughout the manuscript. 

R4.5 Please expand on how perceptions of humanness influence interpersonal relationships (p. 9). If this is important as the authors suggest, the mechanism should be further elucidated.

“It is important that we examine humanness in conjunction with the Stereotype Content Model because perceptions of humanness directly influence interpersonal relationships (44). For instance, people who are dehumanized are more often victims of objectification (43) and aggression , and receive less empathy from others (45). Hence, it is imperative that we understand whether plastic surgery recipients are dehumanized.” p9

R4.6 Please specify whether gender or sex was measured. Male is used as a descriptor throughout, however, typically when gender is measured the referent would be men. Were participants all cisgender?

This was an oversight and we have changed terminology throughout our manuscript to reflect gender (e.g., men, women). 

R4.7 Please clarify why the race category for White includes mixed-race people – is there a disadvantage to having a separate category for mixed-race?

The authors felt that recognising multi-racial participants as members of all racial groups with which they identify would mean participant demographics would be more thoroughly represented in the present study. Further, and perhaps more importantly, we felt it appropriate that participants be given the option to include as many or as few races in their identification as they felt appropriate (i.e. we didn’t want to ‘box’ participants into being either White or mixed-race, if perhaps they identified predominantly as White, but also felt alternate categories applied to them). Finally, we presented a free-text options to participants; if they wanted to identify as mixed-race (or anything else that was not listed), they were able to write this in the text box.

R4.8 The authors should justify why only White women stimuli were chosen, and situate this in a discussion of representation in research. This is particularly relevant given much cosmetic surgery has historically attempted to produce more stereotypically White features.

We have explained our rationale in the method. Further, we have commented on the limited generalisability of our findings in the discussion. 

“White women stimuli were chosen because we intuited that the majority of our sample would be White.” p11

“Speaking further to the generalizability of the study, we note that only White plastic surgery stimuli were used. These findings therefore cannot be generalized to people of color; specifically, we are unable to establish whether plastic surgery recipients who are people of color are subject to the negative plastic surgery effect. Given that plastic surgery has historically attempted to produce more stereotypically White features (e.g., surgeries for the ‘Jewish nose’ or ‘Black nose’), it is especially important that we acknowledge the limited applicability of our findings (28,63).” p24

R4.9 Please provide a citation for the Justice Sensitivity inventory, or clarify if this was developed for the present study. If this was developed for the present study, please provide additional information on scale development procedures that were undertaken.

This was an oversight and we have now included this citation. 

R4.10 Theorizing the link between perceptions of humanness, empathy, and objectification may help to enrich the theoretical setup of the study, particularly given the focus on appearance in the current paper.

See R4.5 

R.411 p.19 – “the present study demonstrates that negative attitudes toward plastic surgery extend specifically to the act of undergoing plastic surgery itself…” The present study cannot demonstrate this as only intention to undergo plastic surgery was assessed; the act of undergoing it was not. Please rephrase.

We have rephrased this section and added examples. 

“We posit that exploring plastic surgery as an intended action (versus a completed action) enables researchers to better assess negative perceptions that pertain exclusively to plastic surgery itself, rather than its associated outcomes (i.e. how recipients look after surgery). In other words, studying perceptions of women planning to have plastic surgery allows us to assess shifts in perception regarding recipient character (e.g., “I don’t condone plastic surgery because it is immoral”) as opposed to regarding recipient appearance (e.g., “I don’t condone plastic surgery because I think it makes women look unappealing”). As such, our study aims to establish exactly how a woman’s decision to undergo plastic surgery shapes others’ perceptions of her, irrespective of her surgical outcomes.”p5

R6.1 Consequences of perception on women's well-being. Although I understand it might be an issue that deserves to be empirically investigated, I do not think that the authors should include a sentence like "we concluded attractive women seeking cosmetic surgery are subject to experience negative psychosocial outcomes" in the abstract. This is too speculative. Same comment for the part at the end of the Low warmth and Competence paragraph. It might be better to have these hypotheses put at the end of the manuscript in the Implications and Future Directions section.

We agree with the reviewer. We have reworded the aforementioned sentence in the abstract to better reflect that the authors can only speculate as to how cosmetic surgery influences psychosocial outcomes. We have also reframed the end of our warmth and competence paragraph. Finally, and most significantly, we have noted in our future directions that these outcomes are speculative only, and that future research is needed to confirm whether the negative plastic surgery effect induces contempt and worsened wellbeing. 

“As such, we contend that attractive women seeking plastic surgery may find themselves experiencing negative psychosocial outcomes; they may be interpersonally and professionally affected by others’ feelings of contempt toward them and may also be less likely to fulfil their basic psychological needs. However, we note that at present these outcomes are purely speculative, and that future research is needed to test these associations.” p4

R6.2 Attractiveness ratings. Although the photographs were pretested in terms of attractiveness, it could have been better to ask participants to rate the targets’ attractiveness, at least for manipulation check. Moreover, it could have been interesting to see whether planning cosmetic surgery could influence attractiveness ratings and, more importantly, whether it could influence the link between attractiveness and personality inferences.

We appreciate the reviewer’s feedback. We agree that including a manipulation check would’ve been beneficial and will apply this suggestion to any future research we conduct using these stimuli. However, given that the Chicago Faces Database has previously undergone rigorous and cross-cultural validity assessments, we remain confident that the manipulation was successful (especially considering that we selected the eight most and eight least attractive faces for our stimuli). We also agree that it would be interesting to assess whether cosmetic surgery influences stimuli attractiveness ratings and the relationship between attractiveness ratings and personality inferences. While we are presently unable to conduct this analysis (given that we did not collect attractiveness ratings), we will include this analysis if we conduct similar research in the future. 

6.3 Regarding the gender effect point raised by one reviewer, I understand the authors do not want to include this point in the manuscript because it was not part of their analyses. Still, they could have included the variable as a covariate, as it is done in many research in which gender could have potential effects.

As previously reported in our first revision letter, with the exception of the single morality outcome, participant gender did not moderate the effect of plastic surgery or participant attractiveness on judgements of the stimuli. Additionally, gender did not have a significant main effect on ratings on our outcome measures. We did not have any theoretical reason to expect gender to play an important role in our study, and given the non-significant impact of gender in our analyses, including it as a covariate would not change any reported results and would, in our opinion, add little to the reported story. We acknowledge that readers may raise a similar question as to the potential impact of gender, and so to address this we have added a sentence to the body of our results summarising the (lack of) gender effects in our study.

“At the request of a reviewer, we also examined whether any of the aforementioned results were moderated by participant gender. We found that only the relationship between plastic surgery condition and morality but not warmth was moderated by gender, such that the relationship was only significant when participants were women. All other analyses were unaffected by gender and thus it was not included in reported models.” p19

6.4 I understood that there were no effects of Justice Sensitivity. But before that, unless I missed it, the authors did not justify the choice of assessing only the Observer's perspective. One could also imagine that the Victim's perspective, for example, could have been a moderator (women feeling less attractive than the target would not want the woman seeking to be more attractive through cosmetic surgery to be even more favored considering that attractiveness can sometimes be a social advantage).

We agree with the reviewer that other subscales of the Justice Sensitivity Inventory (JSI) could’ve acted as a moderator for the negative plastic surgery effect. However, when conceptualising the study, we ultimately felt that the observer subscale of the JSI subsumed both self-oriented and other-oriented feelings of injustice; for example, items such as “I am upset when someone does not get a reward he/she has earned” can reflect either oneself as ‘someone’ or an external individual as ‘someone’. Conversely, items on the victim subscale (i.e. the subscale noted by the reviewer) solely apply to wrongdoing towards oneself. As such, we chose to use the observer subscale because it incorporated several different possible experiences of perceived injustice and we therefore felt it more comprehensive. 

6.5 The authors mentioned the what-is-beautiful-is-good stereotype considered as the typical example of a halo effect. Do the authors consider that the results they found could be considered as an example of a horn effect?

We have included this suggestion.

“These perceptions might be considered a ‘horn’ or ‘negative halo’ effect – a cognitive bias whereby perceptions of an individual are unduly influenced by a single negative trait (19).” p5

Editor 1.1 Certainly, in the previous round of the peer review, one reviewer raised grave concerns about misogynistic views and recommended rejection because of that. Further, they declined my invitation to review your revision. However, aiming for an ethical decision that would be as fair as possible for everyone engaged in this research and the review process, I invited additional reviewers who are experts in feminist studies and the psychological perspectives of plastic surgeries. In my invitation letters, as well as separate emails, I've asked the new (and the original) reviewers to highlight any 'misogynistic' views that the study might express (I had the report and concerns of the original reviewer who raised this issue shared with all of the reviewers). Based on the comments and recommendations of the current reviewers, whom I thank faithfully, I was able to make a well-informed decision on your paper.

We appreciate the editor’s comments. We have pasted below comments from second-round reviewers in support of our manuscript. 

Reviewer #5: As a feminist researcher who has undertaken work on the gendered aspects of cosmetic surgeries, I have been asked by the editor to comment specifically as to whether this paper expresses any misogynistic views. I have read the (revised) paper and the reviewers' comments carefully. This paper absolutely does NOT express any misogynistic views. If anything, it is a woman-focussed paper which demonstrates empathy for the plight of women, who are judged wanting if they do not live up to patriarchal appearance standards, and are also judged harshly if they decide to undergo appearance related surgery. This double standard is the exact point of the paper. As feminist cultural theorist & sociologist Ros Gill points out, “Women are never the right age. We are too young, we’re too old. We are too thin, we’re too fat. We wear too much makeup, we don’t wear enough. We are too flashy in our dress, we don’t take enough care. There isn’t a thing we can do that is right” (2007, p.117) . The authors are attempting to challenge the very misogyny that the reviewer is concerned about. The authors' extremely detailed and thorough reply to the reviewer's concerns on this should allay any concerns on this front, to my mind.

Reviewer #4: Thank you for the opportunity to review the present manuscript: Under the knife: Unfavorable perceptions of women who seek plastic surgery. The manuscript reports on an online experimental study examining perceptions of attractive and unattractive women planning to engage in plastic surgery or control activities. The project is interesting and has sufficient depth, and I applaud the authors’ transparency and engagement with open practices, as well as their considerate and deep engagement with the prior reviews. I believe the manuscript is fundamentally sound, and I do not believe the present manuscript expresses or reifies misogynist views.

---

## [Decision Letter · Decision Letter 3]

27 Jul 2021

PONE-D-21-05038R3

Under the Knife: Unfavorable Perceptions of Women Who Seek Plastic Surgery

PLOS ONE

Dear Dr. Bonell,

Thank you for submitting your manuscript to PLOS ONE. Most of the reviewers have now recommended accepting your revised manuscript for publication. However, before proceeding with this, I invite you to address one correction requested by Reviewer #6.

We look forward to receiving your revised manuscript.

Kind regards,

Ali B. Mahmoud, Ph.D.

Academic Editor

PLOS ONE

Journal Requirements:

Additional Editor Comments (if provided):

Reviewers' comments:

Reviewer's Responses to Questions

**Comments to the Author**

1. If the authors have adequately addressed your comments raised in a previous round of review and you feel that this manuscript is now acceptable for publication, you may indicate that here to bypass the “Comments to the Author” section, enter your conflict of interest statement in the “Confidential to Editor” section, and submit your "Accept" recommendation.

Reviewer #1: All comments have been addressed

Reviewer #2: All comments have been addressed

Reviewer #4: All comments have been addressed

Reviewer #5: All comments have been addressed

Reviewer #6: (No Response)

2. Is the manuscript technically sound, and do the data support the conclusions?

Reviewer #1: Yes

Reviewer #2: Yes

Reviewer #4: Yes

Reviewer #5: Yes

Reviewer #6: Yes

3. Has the statistical analysis been performed appropriately and rigorously? 

Reviewer #1: Yes

Reviewer #2: Yes

Reviewer #4: Yes

Reviewer #5: I Don't Know

Reviewer #6: Yes

4. Have the authors made all data underlying the findings in their manuscript fully available?

Reviewer #1: Yes

Reviewer #2: No

Reviewer #4: Yes

Reviewer #5: Yes

Reviewer #6: Yes

5. Is the manuscript presented in an intelligible fashion and written in standard English?

Reviewer #1: Yes

Reviewer #2: Yes

Reviewer #4: Yes

Reviewer #5: Yes

Reviewer #6: Yes

6. Review Comments to the Author

Reviewer #1: The authors have made substantial improvements to this manuscript, by attending to the previous comments and suggestions.

Reviewer #2: This is an interesting study. The authors have substantially revised the manuscript which makes an interesting contribution to the field.

Reviewer #4: (No Response)

Reviewer #5: (No Response)

Reviewer #6: I am generally satisfied with the way the authros responded to the different comments.

I only have one request. Regarding the Justice Sensitivity point, I appreciate the authors justifying their choice in the response to reviewers section, however, I think it is important that the authors specify in the text and earlier than the Method that they focus only on the Observer's perspoective and why.

The measure created by Schmitt et al. is about all aspects so when one is using only one aspect, I believe it is necessary to justify the decision.

7. PLOS authors have the option to publish the peer review history of their article (what does this mean?). If published, this will include your full peer review and any attached files.

Reviewer #1: No

Reviewer #2: **Yes: **Dr Dieu Hack-Polay

Reviewer #4: No

Reviewer #5: **Yes: **Dr Paula Singleton

Reviewer #6: No

---

## [Author Response · Author response to Decision Letter 3]

28 Jul 2021

6.1 I am generally satisfied with the way the authros responded to the different comments. I only have one request. Regarding the Justice Sensitivity point, I appreciate the authors justifying their choice in the response to reviewers section, however, I think it is important that the authors specify in the text and earlier than the Method that they focus only on the Observer's perspoective and why. The measure created by Schmitt et al. is about all aspects so when one is using only one aspect, I believe it is necessary to justify the decision.

We have added this justification into the body of the manuscript. 

“At the request of a reviewer, we would like to acknowledge that while the Justice Sensitivity Inventory consists of four subscales, we ultimately felt that using solely the Observer subscale best suited the aims of our study. Namely, the Observer subscale subsumes both self-oriented and other-oriented feelings of injustice (i.e., injustices that affect both oneself and others). For example, items such as “I am upset when someone does not get a reward he/she has earned” could represent oneself as ‘someone’ or an external individual as ‘someone’. Conversely, items on other subscales (e.g., the Victim subscale) exclusively measure feelings of injustice towards oneself. As such, we chose to use the Observer subscale because it incorporated several different possible experiences of perceived injustice and we therefore felt it more comprehensive.” p13

---

## [Decision Letter · Decision Letter 4]

25 Aug 2021

Under the Knife: Unfavorable Perceptions of Women Who Seek Plastic Surgery

PONE-D-21-05038R4

Dear Dr. Bonell,

We’re pleased to inform you that your manuscript has been judged scientifically suitable for publication and will be formally accepted for publication once it meets all outstanding technical requirements.

Kind regards,

Ali B. Mahmoud, Ph.D.

Academic Editor

PLOS ONE

Additional Editor Comments (optional):

Reviewers' comments:

Reviewer's Responses to Questions

**Comments to the Author**

1. If the authors have adequately addressed your comments raised in a previous round of review and you feel that this manuscript is now acceptable for publication, you may indicate that here to bypass the “Comments to the Author” section, enter your conflict of interest statement in the “Confidential to Editor” section, and submit your "Accept" recommendation.

Reviewer #6: All comments have been addressed

2. Is the manuscript technically sound, and do the data support the conclusions?

Reviewer #6: Yes

3. Has the statistical analysis been performed appropriately and rigorously? 

Reviewer #6: Yes

4. Have the authors made all data underlying the findings in their manuscript fully available?

Reviewer #6: Yes

5. Is the manuscript presented in an intelligible fashion and written in standard English?

Reviewer #6: Yes

6. Review Comments to the Author

Reviewer #6: (No Response)

7. PLOS authors have the option to publish the peer review history of their article (what does this mean?). If published, this will include your full peer review and any attached files.

Reviewer #6: No

---

## [Editor Report · Acceptance letter]

27 Aug 2021

PONE-D-21-05038R4 

Under the knife: Unfavorable perceptions of women who seek plastic surgery 

Dear Dr. Bonell:

I'm pleased to inform you that your manuscript has been deemed suitable for publication in PLOS ONE. Congratulations! Your manuscript is now with our production department. 

Kind regards, 

on behalf of

Dr. Ali B. Mahmoud 

Academic Editor

PLOS ONE